

# Probabilistic inference of ecohydrological parameters using observations from point to satellite scales

Maoya Bassiouni[1], Chad W Higgins[1], Christopher J Still[2], and Stephen P Good[1]

[1]Department of Biological and Ecological Engineering, Oregon State University, Corvallis, OR 97333, USA
5  [2]College of Forestry, Oregon State University, Corvallis, OR 97333, USA

*Correspondence to*: Maoya Bassiouni (bassioum@oregonstate.edu)

**Abstract.** Ecohydrological parameters that describe vegetation controls on soil moisture dynamics are not easy to measure at hydrologically meaningful scales and site-specific values are rarely available. We hypothesize that sufficient information 10  required to determine these ecohydrological parameters is encoded in empirical probability density functions (pdfs) of soil saturation, and that this information can be extracted through inverse modeling of the commonly used stochastic soil water balance. We developed a generalizable Bayesian inference approach to estimate soil saturation thresholds at which plants control soil water losses, based only on soil texture, rainfall and soil moisture data at point, footprint, and satellite scales. The optimal analytical soil saturation pdfs were statistically consistent with empirical pdfs and parameter uncertainties were on 15  average under 10 %. Parameter estimates were most constrained for scales and locations at which soil water dynamics are more sensitive to the fitted ecohydrological parameters of interest. The algorithm convergence was most successful and the best goodness-of-fit statistics were obtained at the satellite scale. Robust and accurate results were obtained with as little as 75 daily observations randomly sampled from the full records, demonstrating the advantage of analyzing soil saturation pdfs instead of time series. A sensitivity analysis showed that estimates of soil saturation thresholds at which plants control soil 20  water losses were not sensitive to soil depth and near-surface observations are valuable to characterize ecohydrological factors driving soil water dynamics at ecosystem scales.  This work combined modeling and empirical approaches in ecohydrology and provided a simple framework to obtain analytical descriptions of soil moisture dynamics at a range of spatial scales that are consistent with soil moisture observations.

## 1 Introduction

25  The movement of water from soils, through plants, and back the atmosphere via transpiration, is a critical component of local and global hydrologic cycles, and is the largest surface-to-atmosphere water pathway (Good et al., 2015). A realistic analytical description of soil moisture dynamics is key to understanding ecohydrological processes that regulate the productivity of natural and managed ecosystems. Rodrigues-Iturbe et al. (1999) introduced a conceptually simple framework using a bucket model of soil-column hydrology forced with stochastic precipitation inputs, where soil water losses are only a 30  function of soil saturation. Given this ecohydrological framework, the probability density function (pdf) of soil moisture and the mean components of the soil water balance are analytically derived and depend on simple abiotic characteristics such as



average climate and soil texture, and biotic characteristics including soil saturation thresholds at which vegetation can influence soil water losses. However, the shapes of analytical soil moisture pdfs are generally not consistent with observations when literature values for model parameters are used (Miller et al., 2007). Analytical pdfs have never been directly compared to empirical pdfs derived from measurements beyond the point scale. Observation networks provide freely

available point scale, spatially integrated soil moisture observations, while remotely sensed soil moisture observations are available through satellite products. These data sources create an opportunity to: 1) evaluate whether analytical soil saturation pdfs are consistent with observations across a range of scales; and 2) determine ecohydrological parameters relevant to each scale.

Estimates of ecohydrological parameters are relevant to a large range of applications for which the stochastic soil water balance framework has been used and adapted, including: the effects of climate, soil and vegetation on soil moisture dynamics (Laio et al, 2001a; Rodrigues-Iturbe et al., 2001; Porporato et al., 2004), ecohydrological factors driving spatial and structural characteristics of vegetation (Caylor et al., 2005; Manfreda et al., 2017), soil salinization dynamics (Suweis et al., 2010), biological soil crusts (Whitney et al., 2017), vegetation stress, optimum plant water use strategies and plant hydraulic

failure (Laio et al., 2001b; Manzoni et al. 2014; Feng et al., 2017), vertical root distributions (Laio et al., 2006), plant pathogen risk (Thomspon et al., 2013), streamflow persistence in seasonally dry landscapes (Dralle et al., 2016), and soil water balance partitioning (Good et al., 2014 ; Good et al.,   http://rdcu.be/yqW7). A survey of close to 400 echoydrology publications found that 40% relied heavily on simulation, rarely integrated empirical measurements, and were almost never coupled with experimental studies, suggesting a critical need to combine modeling and empirical approaches in

echohydrology (King and Caylor, 2011). Few studies have directly confronted the governing equations of the stochastic soil water balance model with observed soil moisture data and fewer have attempted to optimize model parameters to best fit soil moisture observations. Miller et al., (2007) calibrated soil moisture pdfs to project vegetation stress in a changing climate. Chen et al., (2008) related evapotranspiration observations at the stand scale to soil moisture values using a Bayesian inversion approach, and Volo et al., (2014) calibrated the soil moisture loss curve to investigate effects of irrigation

scheduling and precipitation on soil moisture dynamics and plant stress. The functional form of the soil moisture losses was approximated using conditionally averaged precipitation (Salvucci, 2001; Saleem and Salvucci, 2002) and remotely sensed data (Tuttle and Salvucci, 2014). The time scale of soil moisture dry downs, derived from the soil moisture loss equations, were parameterized using evapotranspiration measured at micro-meteorological stations (Teuling et al., 2006) and space-born near-surface soil moisture observations (McColl et al., 2017). These studies indicate that the ecohydrological soil water

balance framework is consistent with ground and remotely sensed measurements.

This study expands upon previous work and presents sensitivity tests to generalize the direct inference of ecohydrological parameters and associated uncertainty, from observed soil moisture pdfs at a range of scales. We hypothesize that key information required to determine the ecohydrological factors driving soil moisture dynamics is encoded in empirical soil



saturation pdfs, and that this information can be extracted by calculating the inverse of the commonly used stochastic soil water balance. Non-biological controlling factors on the soil water balance can generally be assessed from readily available data, including site measurements, regionalized maps, and satellite observations. Vegetation controls on soil water dynamics are largely unknown and difficult to measure at hydrologically meaningful scales (Li et. al., 2017). We thus focused on estimating parameters that are not generally observable, in particular the soil saturation thresholds at which vegetation controls soil water losses, through an inverse modelling approach and using data that are commonly collected at environmental monitoring sites. Analysis of soil saturation pdfs is a more robust and integrated approach to investigate ecohydrological factors of soil water dynamics than time series analysis. Soil saturation pdfs are less sensitive to the many sources of uncertainty and common gaps in soil moisture observations and do not require high quality co-located and concurrent hydrologic measurements that are often lacking. A number of studies have combined inverse modeling approaches with ground and remotely sensed soil moisture data to successfully extract meaningful hydrologic information (Xu et al., 2006; Miller et al, 2007; Chen et al., 2008; Volo et al., 2014; Wang et al., 2016; Baldwin et al., 2017). In particular, Bayesian inference methods are effective in relating prior pdfs of observations to posterior estimates of model parameters (Xu et al., 2006; Chen et al., 2008; Baldwin et al., 2017). The soil water balance model provides a direct analytical equation for soil moisture pdfs that is convenient to use with the Bayesian paradigm because it is a low parameter model with few data inputs. In this study, we developed a Bayesian inversion approach to directly estimate soil water balance model parameters that best fit soil moisture pdfs derived from observations at point, footprint, and satellite scales. The Bayesian approach quantifies the interference uncertainty directly and improves upon the work of Miller et al. (2007), which used a least-squares approach to calibrate soil saturation pdfs.

Parameters that are representative of larger scale observations are necessary to characterize ecohydrological processes at ecosystem scales and are more relevant to ecohydrological modelling. In addition, the resulting inference framework provides a means to compare the sensitivity of soil moisture dynamics at varying scales to simple ecohydrological parameters. The generalization of the proposed approach was evaluated using co-located and concurrent soil moisture observations at the point, footprint, and satellite scales. To our knowledge, this is the first study to infer parameters for the analytical model of soil saturation pdfs for scales beyond point observations. We sought to evaluate 4 key questions necessary to generalize the inference of ecohydrological parameters: (1) What is the minimum level of model complexity needed to obtain consistent analytical and empirical soil saturation pdfs, and which parameters can be inferred with the most certainty? (2) Are ecohydrological parameter estimates sensitive to the soil moisture sensing depth, and can we assume a homogenous soil column of a depth greater than the sensing depth? (3) What is the minimum amount of data necessary to estimate ecohydrological parameters through a Bayesian inversion of soil saturation pdfs? (4) At which scales and sites are ecohydrological parameter estimates most accurate and pertinent?




The goal of this study was to confront empirical soil moisture pdfs derived from point-, footprint-, and satellite-scale observations to a commonly used analytical model. We demonstrate the use of a Bayesian inversion framework to infer the ecohydrological parameters of a simple stochastic soil water balance model that best fit empirical soil moisture pdfs. We first present data sources, define the analytical model for soil moisture pdfs including parameter assumptions, and detail the
algorithm used in the Bayesian inversion. Then, we present a summary of the goodness of fit of optimal analytical soil moisture pdfs and estimated parameter uncertainty for a range of sensitivity tests. Results of sensitivity tests were used to define criteria for a generalization of the presented approach to future applications. Finally, we discuss the potential of the approach to provide a simple means to investigate variability in ecohydrological controlling factors at varying spatial scales. This work combines modelling and empirical approaches in echohydrology to provide more realistic analytical descriptions
of soil moisture dynamics. Estimates of ecohydrological parameters that are consistent with observed soil moisture pdfs, from point to ecosystem scales, are needed to better characterize site-specific ecohydrological processes.

## 2. Data and Methods

### 2.1 Data analysed

Daily soil moisture observations from three data products at three different spatial scales were used in this study. Point-scale
soil moisture at 10 cm depth was taken from the FLUXNET2015 data product (http://fluxnet.fluxdata.org/data/fluxnet2015-dataset/). Footprint-scale soil moisture was taken from the Cosmic-ray Soil Moisture Observing System (COSMOS) (http://cosmos.hwr.arizona.edu/Probes/probelist.html). The COSMOS soil moisture footprint measures soil moisture at an average depth of 20 cm with a radius ranging from 130 to 240 m, depending on site characteristics (Köhli et al., 2015). Near-surface soil moisture observations at a spatial resolution of 0.25˚ were taken from the European Space Agency's (ESA)
Climate change Initiative (CCI) project. The combined soil moisture product (ECV-SM, version 0.2.2) that merges soil moisture retrievals from four passive (SMMR, SMM/I, TMI, and ASMR-E) and two active (AMI and ASCAT) coarse resolution microwave sensors was used (Liu et al., 2011; Liu et al., 2012; Wagner, 2012). Although the ECV-SM sensing depth is less than 5 centimeters, it has been shown to have a close relation to ground-based observations of soil moisture in the upper 10 centimeters (Dorigo et al., 2015). Daily rainfall time series were compiled from the FLUXNET2015 dataset for
the point-and footprint-scale analysis, and the National Aeronautics and Space Administration's (NASA) Tropical Rainfall Measuring Mission (TRMM) dataset (Huffman et al., 2007) for the satellite-scale analysis. The growing season of May to September 2012 was selected for analysis because concurrent rainfall and soil moisture observations for each soil moisture and rainfall data product were available during this time period for a maximum number of sites.

In total, 4 sites with data available during April to September 2012 were selected for this analysis (Table 1). Selected sites span a range of land cover types including crop and grasslands, oak savanna, deciduous forest and pine forest. For each site, the dominant soil texture of the upper soil layer was determined from the Harmonized World Soil Database (HWSD) (version





1.2) (FAO/IIASA/ISRIC/ISS-CAS/JRC, 2012). Soil porosity values, derived from the HWSD available as ancillary data through the ESA-CCI data product were used for the satellite scale analysis. For point- and footprint-scale data products, the maximum soil moisture observation during the year 2012 was used as a site-specific soil porosity estimate. Soil porosity for each site was applied to compute the relative soil moisture content or soil saturation ($0 \leq s \leq 1$) from each observed soil

moisture value. Soil saturation and rainfall data at each scale and for each site during the selected analysis period are presented in Fig. 1. All sites had 183 daily point-, and footprint-scale observations and between 109 and 153 daily satellite-scale observations. We consider that during the selected analysis period May to September 2012 the steady state assumption is met.

## 2.2 Analytical model for soil saturation probability density functions (pdfs)

### 2.2.1 Model definition

The framework used in this study is based on a standard bucket model of soil column hydrology at a point forced with stochastic precipitation inputs and in which soil water losses are a function of soil saturation. We follow the simple formulation of soil water losses in Laio et al. (2001a) and apply the associated analytical formulation for the pdf detailed

below. However, the methodology described in Sect. 2.3 can be customized to characterize site-specific parameters and test consistency between observed and analytical soil saturation pdfs for any application or adaptation of the stochastic ecohydrological framework.

The soil water balance model is defined at a point scale and a daily time scale, for a soil with porosity $n$ and depth $Z$, and

assumes soil saturation is uniform in the rooting zone. Rainfall, the only input to the soil water balance, is treated as a Poisson process characterized by an average event frequency, $\lambda$, and average event intensity, $\alpha$. For simplification, we assume that the rainfall applied is equal to the amount reaching the ground surface and do not account for rainfall intercepted by vegetation. The daily soil water balance is written as the difference between $\varphi$, the rate of infiltration from rainfall and $\chi$, the rate of soil moisture losses:

$$nZ \frac{ds(t)}{dt} = \varphi[s(t); t] - \chi[s(t)] \tag{1}$$

$\varphi[s(t); t]$ is a stochastic process controlled by rainfall and is also a state-dependent process, because excess rainfall relative to available soil storage is converted to surface runoff. $\chi[s(t)]$, the soil moisture loss curve, is summarized in Fig. 2a and includes leakage losses due to gravity and evapotranspiration and is described in stages determined by five soil saturation thresholds (Laio et al., 2001a). These stages are: (1) the saturation point ($s = 1$), at which all pores are filled with water; (2)

the field capacity ($s_{fc}$), at which soil-gravity drainage becomes negligible compared to evaporation; (3) the point of incipient stomata closure ($s^*$), at which plants begin to reduce transpiration from water stress; (4) the wilting point ($s_w$), at which plants cease to transpire; and (5) the hydroscopic point ($s_h$), at which water is bound to the soil matrix. Soil water losses are





controlled by physical soil properties for saturation states above $s_{fc}$. The rate of leakage due to gravity is assumed maximum when the soil is saturated ($K_s$) and decays exponentially to a value of 0 at $s_{fc}$ (Brooks and Corey, 1964). Soil water losses are controlled by micro-meteorological conditions for saturation states between $s_{fc}$ and $s^*$. The rate of evapotranspiration is assumed to occur at a constant maximum rate ($E_{max}$). Soil water losses are controlled primarily by vegetation for saturation

5    states between $s^*$ and $s_w$. Plants close their stomata in response to soil water deficits that drive leaf water potential gradients, as well as to atmospheric vapor pressure deficits, and evapotranspiration decreases linearly from $E_{max}$ to $E_w$ at $s_w$. Soil water losses are controlled by soil diffusivity for soil saturation states below $s_w$, and soil evaporation decreases linearly from $E_w$ to 0 at $s_h$. Soil water losses are negligible for soil saturation states below $s_h$. The piece-wise linear relation between soil saturation and evapotranspiration is a simplifying assumption commonly used is soil water balance models.

For this simplified theoretical description of the soil water loss curve and stochastic rainfall forcing, the analytical solution of the steady-state probability distributions of soil saturation, $p(s)$ given by Laio et al. (2001a) is:

$$
p(s) = \begin{cases}
0, & 0 < s \le s_h, \\[2mm]
\frac{C}{\eta_w}\left(\frac{s-s_h}{s_w-s_h}\right)^{\frac{\lambda(s_w-s_h)}{\eta_w}-1} e^{-\gamma s}, & s_h < s \le s_w, \\[3mm]
\frac{C}{\eta_w}\left[1+\left(\frac{\eta}{\eta_w}-1\right)\left(\frac{s-s_w}{s^*-s_w}\right)\right]^{\frac{\lambda(s^*-s_w)}{\eta-\eta_w}-1} e^{-\gamma s}, & s_w < s \le s^*, \\[3mm]
\frac{C}{\eta} e^{-\gamma s + \frac{\lambda}{\eta}(s-s^*)}\frac{\eta}{\eta_w}^{\frac{\lambda(s^*-s_w)}{\eta-\eta_w}}, & s^* < s \le s_{fc}, \\[3mm]
\frac{C}{\eta} e^{-(\beta+\gamma)s+\beta s_{fc}}\left(\frac{\eta e^{\beta s}}{(\eta-m)e^{\beta s_{fc}}+me^{\beta s}}\right)^{\frac{\lambda}{\beta(\eta-m)}+1}\frac{\eta}{\eta_w}^{\frac{\lambda(s^*-s_w)}{\eta-\eta_w}}e^{\frac{\lambda}{\eta}(s_{fc}-s^*)}, & s_{fc} < s \le 1,
\end{cases}
\tag{2}
$$

where

15    $\frac{1}{\gamma} = \frac{\alpha}{nZ}$,

$\eta_w = \frac{E_w}{nZ}$,

$\eta = \frac{E_{max}}{nZ}$,

$m = \frac{K_s}{nZ\left(e^{\beta(1-s_{fc})}-1\right)}$,

$\beta = 2b-4$.

20    where $b$, is an experimentally determined parameter used in the Clapp and Hornberger, (1978) soil water retention curve and the constant $C$ can be obtained numerically to ensure the integral of $p(s)$ is equal to 1. This framework was derived under the assumption of steady state, wherein parameters are constant for a given period of time.



### 2.2.2 Climate, soil and vegetation parameter characterization

The rainfall characteristics ($\lambda$ and $\alpha$) and physical soil parameters ($s_{fc}$, $s_h$, $K_s$, and $b$) used in Eq. (2) are based on readily available data. We chose values based on our best estimates of the driving climate and physical soil controls on the soil water balance. We thus focused on estimating the ecohydrological parameters $s^*$, $s_w$, $E_{max}$, and $E_w$, which describe vegetation

5 controls on soil water losses and are not easily observable. We acknowledge that the pre-defined rainfall characteristics and physical soil parameters based on observations or literature values may not be perfectly representative of the processes at each location or scale and could create biases and uncertainties in our fitted parameters of interest.

Rainfall characteristics $\lambda$ and $\alpha$ were calculated for each site from the FLUXNET2015 and TRMM rainfall records during the

10 selected 2012 growing season following Rodriguez-Iturbe et al. (1984). The FLUXNET2015 rainfall characteristics were used for the point- and footprint-scale analysis, while the TRMM rainfall characteristics were used for the satellite-scale analysis. Physical soil characteristics, $s_h$, $K_s$, and $b$ were taken from Rawls et al. (1982) and are listed for each site in Table 1. To be most consistent with the assumption that drainage losses are generally insignificant compared to evapotranspiration losses the day following a rain event, $s_{fc}$ was estimated from each soil saturation record and listed in Table 1. All days in the

15 2012 record immediately following an observed increase in soil saturation were identified and $s_{fc}$ was estimated as the 95th percentile of the soil saturation values on these selected days. The soil saturation pdfs in this study generally indicate that soil moisture states below $s_w$ and above $s^*$ are rare, therefore we do not expect the pre-defined values for $s_{fc}$, $s_h$ and $K_s$ to significantly affect results. The framework we present thus considers 4 unknown soil water balance parameters, $s^*$, $s_w$, $E_{max}$ and $E_w$. Our goal is estimate these parameters, as defined over the following intervals:

$$\begin{cases} s_h \leq s^* \leq s_{fc}, \\ s_h \leq s_w \leq s_{fc}, \\ 0 \leq E_{max} \leq 10, \\ 0 \leq E_w \leq 5 \end{cases} \tag{3}$$

where 10 and 5 mm day$^{-1}$ are the pre-defined upper possible bounds for potential evapotranspiration and actual evapotranspiration at the wilting point. Estimates of $s^*$ and $s_w$ can be converted to soil matrix potential if soil water retention parameters are well known. The Clapp and Hornberger, (1978) soil water retention curve is highly non-linear and estimates of soil water potential at which stomata fully are open or closed were not evaluated in this study.

### 2.2.3 Model complexity descriptions

We considered the following 4 levels of complexity for the soil water loss curve model:

    (i)   evapotranspiration decreases linearly from $E_{max}$ to 0 between $s_{fc}$ and $s_h$,

    (ii)  evapotranspiration is maximum between $s_{fc}$ and $s^*$, then decreases linearly from $E_{max}$ to 0 between $s^*$ and $s_h$,





(iii) evapotranspiration decreases linearly from $E_{max}$ to $E_w$ between $s_{fc}$ and $s_w$, then decreases linearly from $E_w$ to 0 between $s_w$ and $s_h$. We also test a variation of model (iii), assuming $E_w = 0.05E_{max}$ and call this model (iii')

(iv) evapotranspiration is maximum between $s_{fc}$ and $s^*$, decreases linearly from $E_{max}$ to $E_w$ between $s^*$ and $s_w$, then decreases linearly from $E_w$ to 0 between $s_w$ and $s_h$. We also test a variation of model (iv), assuming $E_w = 0.05E_{max}$ and call this model (iv').

The simplest model (i) has one unknown parameter, and the most complex model (iv), equivalent to the Laio et al., 2001 model, has 4 unknown parameters. We used the simplifying relation $E_w = 0.05E_{max}$ to reduce the number of model parameters in models (iii') and (iv'). For iii' and iv' a range of $E_w/E_{max}$ fractions were tested (not shown), and although overall the method was not sensitive to this parameter, 0.05 was selected to provide converging results with low uncertainty. Models (i) – (iv) are defined in Table 2 and illustrated in Fig. 2. We evaluated models (i) – (iv) to determine which level of complexity is consistent with soil moisture observations and which parameters could be estimated with most certainty.

## 2.3 Bayesian inversion approach

### 2.3.1 Application of the Bayes theorem

Bayes' theorem, Eq. (4) is used to relate $p(S)$, the empirical soil saturation pdf of $j = [1, ..., m]$ soil saturation observations ($s_j$) and the analytical soil saturation pdfs in Eq. (2), derived from the simple soil water balance model in Eq. (1), with 4 unknown soil water balance parameters $\theta = [s^*, s_w, E_{max}, E_w]$.

$$p(\theta|S) = \frac{p(S|\theta)\, p(\theta)}{p(S)} \tag{4}$$

The posterior distribution, $p(\theta|S)$, is the solution of the inverse problem and describes the probability of model parameters $\theta$ given the set $S = [s_1, s_2, ... s_m]$ of soil saturation observations. Assuming uninformed prior knowledge, the prior distribution of model parameters $\theta$, $p(\theta)$, are defined by uniform distributions over the intervals in Eq. (3). The conditional probability of observations $S$ given model parameters $\theta$, $p(S|\theta)$, is the likelihood function of model parameters $\theta$.

### 2.3.2 Parameter estimation and evaluation

The Metropolis-Hasting Markov chain Monte Carlo (MH-MCMC) technique is used to estimate the posterior distribution of $p(\theta|S)$ by drawing random model samples $\theta_i$ from $p(\theta)$ and evaluating $p(S|\theta_i)$ (Metropolis et al., 1953; Hastings, 1970; Xu et al., 2006). The likelihood function of a model $i$, $p(S|\theta_i)$ defined by

$$p(S|\theta_i) = \prod_{j=1}^{m} p(s_j|\theta_i) \tag{5}$$

where $p(s_j|\theta_i)$ is the probability of observation $s_j$ given the model in Eq. (2) using parameters $\theta_i$.



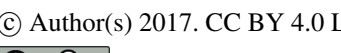

The MH-MCMC technique converges to a stationary distribution according to the ergodicity theorem in Markov chain theory. The sampling algorithm consists of repeating two steps: (1) a proposing step, in which, the algorithm generates a new model $\theta_i'$ using a random function that is symmetric about the previously accepted model $\theta_i$, and (2) a moving step, in which, $\theta_i'$ is tested against the Metropolis criterion $(a)$ to estimate if it should be accepted or rejected.

$$a = \frac{p(S|\theta_i')}{p(S|\theta_i)} \tag{6}$$

If $a > 1$, then $\theta_i$ is accepted and $\theta_{i+1} = \theta_i'$ is used for the next sample. If $a < 1$, a random number $p_* \in [0,1]$ is drawn from a uniform distribution and compared to $a$. If $p_* < a$, then $\theta_i'$ is accepted and $\theta_{i+1} = \theta_i'$ is used for the next sample. If $p_* > a$, $\theta_i'$ is rejected and $\theta_{i+1} = \theta_i$ is used for the next sample. If $\theta_i'$ is an inconsistent model in which the soil saturation thresholds $(s_w, s_*)$ are ranked incorrectly or any of the soil water balance parameters $(s^*, s_w, E_{max}$ and $E_w)$ are outside of their defined physical bounds, the model likelihood is 0 and $\theta_i'$ is never accepted. In this study, the log-likelihood was more convenient to compute than the likelihood. The symmetric function used in the proposing step was a Gaussian distribution with a mean value equal to the accepted model $\theta_i$ and a standard deviation of 1 percent of interval range for which each parameter is defined in Eq. (3).

The value of the standard deviation of each model parameter was set after a number of test runs to generally ensure an acceptance rate between 20 and 50% (Robert and Rosenthal, 1998). Statistics of the estimated parameters in $\theta$ are obtained from the union of 5 run samples of 20 thousand simulations each. The burn-in period is the number of simulations after which the running mean and standard deviation are stabilized. We considered a burn-in period of 10 thousand simulations, which were discarded for each run sample. If the acceptance rate of a run sample is below 5% or greater than 80% after the burn-in period, the run was discard and we concluded that the algorithm converged to a local minimum that may be physically impossible. If more than 10 run sample were performed without retaining 5 run samples, we concluded that the soil saturation record did not contain enough information to estimate $\theta$. Convergence was evaluated by the Gelman-Rubin (GR) diagnostic (Gelman and Rubin, 1992) on the final 5 run samples. The GR diagnostic determines that the algorithm reaches convergence when the within-run variability $(\sigma_w)$ is roughly equal to the between-run variability $(\sigma_b)$, i.e. $\sigma_w/\sigma_b$ approaches 1. For records that obtain 5 converging run samples, the mean and standard deviation of each parameter from the total of 50 thousand simulations of $\theta$ were computed. A mean analytical model of soil saturation pdf was determined by applying Eq. (3) with the mean values of the 50 thousand posteriori parameter estimates. The Kolmogorov-Smirnov statistic and Quantile-Quantile plots were used to evaluate the consistency of the mean analytical model and the empirical soil saturation pdfs. Calculations in this study relied on supercomputer resources through the Extreme Science and Engineering Discovery Environment (XSEDE) (Towns et al., 2014). Custom scripts in the Python computing language associated with this analysis are available through a gitHub repository (ciation TBD).





### 2.4 Description of sensitivity tests

This study investigates questions of model complexity, uncertainty in parameter estimation, data availability, and scales of applicability through the following four levels of sensitivity analysis. Each level of analysis was repeated 10 times using soil moisture records at each scale and site to obtain robust median results.

(1) We applied the inversion framework to variations of the analytical model for soil saturation pdfs (Eq. 3) of increasing complexity from one to four unknown parameters (Table 1, Fig. 2). We determined which parameters can be estimated with acceptable certainty and if more parsimonious analytical models for soil saturation pdfs are consistent with empirical pdfs and may be more robust to use.

(2) We performed the model inversion with a range of rooting depths between 5 cm and 1 m. We determined whether
the approach using near-surface soil saturation observations can evaluate the soil water balance over a range of deeper rooting depths $Z$. We tested the assumption of a homogenous soil column and evaluated the sensitivity of the rooting depth on the estimation of soil water balance parameters. This analysis also determined whether it is necessary to input the exact soil moisture sensing depth, which is often unknown for larger-scale observations, to accurately perform the model inversion.

(3) We performed the model inversion with subsets of each soil saturation record by randomly resampling fractions of the data down to 20 % of the record (April and September 2012). We determined the number of data points necessary to infer converging model parameters that best match observations and which data availability criteria influence the convergence and accuracy of the model inversion.

(4) We compared co-located parameter estimates and their uncertainty at a range of scales for each site by integrating
findings from the above levels of analysis. We determine whether the soil saturation pdf model inversion framework is applicable to point, footprint, and satellite scale observations and whether inferred parameters can be appropriate for ecohydrological modelling at all scales and locations. Co-located and concurrent soil saturation pdfs at a range of scales and their associated model parameter estimates were used to understand whether average ecohydrological parameters vary with scale.

## 4. Results and discussion

For each of the 4 selected locations, optimal analytical soil saturation pdfs consistent with empirical pdfs derived from soil saturation observations were successfully obtained through the Bayesian inversion framework and using a MH-MCMC algorithm. Figure 3 presents a comparison between empirical and analytical pdfs with associated quantile-quantile plots for point, footprint, and satellite scales at the 4 study sites. The (iv') model variation was used (see Sect. 4.2) with Z equal to the
sensing depths of 10, 20, and 5 cm for the point, footprint, and satellite scales, respectively. The Kolmogorov-Smirnov statistic ranged from 0.05 to 0.11; associated p-values were greater than 5-percent statistical significance except for the point and footprint scale results at US-Ton, which had a p-value of 0.02. Posteriori probability distributions of soil water balance



parameters $(s_w, s^*, E_{max})$ were overall well constrained. The coefficient of variation of posteriori distributions were on average 7 %, and ranged between 1 and 23 % for all sites and scales.

### 4.1 Level of model complexity

For each spatial scale and site, the 6 model variations in Table 2 were each inversed using 8 Z values ranging from 5 cm to 40 cm, with 10 repeats for each case. Results were used to determine how many and which soil water loss parameters can be inferred from soil saturation pdfs with most certainty. Only the converging model inversions among the 80 model-site-scale combinations were retained and their median results were summarized in Fig. 4. The most successful parameter estimations were obtained using model variations (iii'), (i), and (iv') with 97, 94, and 85 percent converging results, respectively, compared to model variations (ii), (iii), and (iv) with 52, 44, and 42 percent converging results. The model goodness of fit generally increased with model complexity. The average Kolmogorov-Smirnov statistic for all model (iv') results was 0.08 with 64 percent that were statistically significant, compared to 0.2, with only 11 percent that were statistically significant for model variation (i). Soil saturation pdfs were therefore more accurately described if $s_w$ and $s^*$ soil threshold parameters are included in the soil water loss equation. Convergence and goodness of fit results were generally better for model variation (iii') than (ii), suggesting that $s_w$ was more important in the analytical equation for soil saturation pdfs and soil water loss equations than $s^*$. The mean coefficient of variation of the posteriori parameter values, converging cases combined, were, 5, 6, 9, and 30 percent for $s_w$, $s^*$, $E_{max}$, and $E_w$, respectively. The coefficient of variation of *a posteriori* values of a parameter was directly related to how sensitive the theoretical shape of soil saturation is to that parameter and inversely related to how accurately that parameter can be estimated. Models (iii) and (iv), in which $E_w$ was an unknown were the least successful. Information may be missing to accurately estimate $E_w$ for most sites. Results indicate that the goodness of fit of soil saturation pdfs and values of other fitted parameters were not very sensitive to the exact value of $E_w$. The simplifying relation $E_w = 0.05E_{max}$ prevented equifinality in the analytical equation for soil saturation pdfs and reduced uncertainty in the inference of the other soil water loss parameters. We conclude that all parameters except $E_w$ can be inferred with high certainty. Given the data available in this study, model (iv') is the most appropriate, and only this model variation was used to obtain results described in the following paragraphs.

### 4.2 Soil depth sensitivity

For each spatial scale and site, the (iv') model variation was inverted for Z values ranging from 5 cm to 1 m, with 10 repeats for each case. Results were used to determine whether the inference of soil water balance parameters was sensitive to the sensing depth and if the resulting analytical model for soil saturation pdfs can be relevant to evaluate the soil water balance for a greater soil depth. Only the converging model inversions among the 10 site-scale-depth combinations were retained and



their median results were summarized in Fig. 5. The soil depth used in the analytical equation for soil saturation pdfs didn't generally impact model inference, parameter uncertainty, and goodness of fit. The influence of soil depth decreased as scale increased and was lowest at satellite scales. For the two drier sites (US-Ton and US-Me2), acceptable results were only obtained for shallower soil depths (below 40 cm) at the point and footprint scales. The Kolmogorov-Smirnov statistic was

generally optimal for Z values between 15 and 60 cm. Estimated values for $s_w$ and $s^*$ were generally not sensitive to the considered soil depth and remained relatively stable. It is expected that $E_{max}$ would scale with soil depth to account for daily soil water losses from a deeper soil reservoir. Although it is conceptually more consistent to consider the actual sensing depth to infer a best-fit model for soil saturation pdfs, we conclude that the model used was not very sensitive to soil depth and methods can be applied with Z values around actual rooting depths.These findings are consistent with discussion related to

the sensitivity of the mean soil water components to soil depth in Laio et al. (2001), and indicate that near-surface soil moisture can be used reliably to relate inferred model results to soil water dynamics in the rooting zone. These results also indicate that parameter estimates are not sensitive to the soil moisture sensing depth. This is particularly relevant to larger scale soil moisture observations, particularly from satellites, when the sensing depth is not accurately quantified.

**4.3 Data availability**

For each spatial scale and site, the (iv') model was inversed with Z values ranging from 5 cm to 40 cm, using random subsamples of 100 to 20 percent of the April – September, 2012 record, and with 10 repeats for each case. Results were used to determine the minimum number of observations necessary to obtain an accurate model inversion. Only the converging model inversions among the 80 subsampled site-scale combinations were retained and their median results were summarized in Fig. 6. For all sites and scales the number of observations did not significantly impact model inference. Although the

Kolmogorov-Smirnov statistic, parameter uncertainty and number of non-converging results increased slightly with decreasing number of observations, acceptable results were always obtained and parameter values were stable. The Kolmogorov-Smirnov statistic generally indicated that best agreement between analytical and empirical pdfs were obtained with over 75 observations. For subsamples with more than 75 daily observations the average fraction of converging model inversions was 85 %. Model parameter values were not sensitive to the number of observations used. Results indicate that

there wasn't a limiting number of observations necessary to obtain accurate parameter estimates when the mean and standard deviation of the randomly selected observations were most consistent with the full record and therefore representative of the rainfall characteristics. The MH-MCMC algorithm was also more likely to not reach convergence when the pdfs of the subsample and the full record were inconsistent.

**4.4 Site and scale considerations**

Soil saturation states at drier sites may be more controlled by soil water loss parameters, while soil saturation states at wetter sites may be more controlled by rainfall characteristics. Model inference at wetter sites, where the rainfall characteristics are

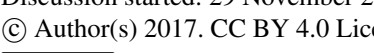



known in this study, is therefore more successful than at dry sites. Although modeled pdfs are in good agreement with empirical pdfs for the wetter sites (US-ARM and US-MMS), parameter estimates can have higher uncertainty because the shape of the soil saturation pdfs are less sensitive to the soil water loss equation parameters. For the drier sites (US-Ton and US-Me2), the shapes of the soil saturation pdfs are more sensitive to the soil water loss equation parameters, the range of

plausible parameters is reduced, and uncertainty can be lower. The MH-MCMC algorithm can be adjusted, if more information were available, to account for the smaller parameter space at drier sites and improve the efficiency of the algorithm. In this study, we discarded results for which the MH-MCMC efficiency was lower than 5 % or greater than 80%.

Similarly, soil saturation states representing larger spatial scales are less sensitive to specific site characteristics, and this
study showed model inference at the satellite scale was generally more successful, while parameter uncertainty was greater than for point and footprint scales. Overall a greater number of analytical pdfs were statistically equal (with 95 % confidence) to empirical pdfs derived from satellite data than from ground-based data. Estimates of larger scale soil water balance parameters are more relevant to regional ecohydrological dynamics. Differences in parameter estimates between scales within a site may be associated with differences in soil texture properties, such as porosity and field capacity, that were determined
separately for each record. Figure 3 also shows that co-located and concurrent soil saturation pdfs are different at each scale and suggest variability in soil water dynamics that are inferred at each scale. Differences in controlling processes between scales were specifically determined from the model inversion for each scale, and provided robust scale-specific parameters for ecohydrological modelling. This study also demonstrated the benefits of analyzing soil saturation pdfs verses time series to understand soil water dynamics, and in particular the appropriateness of the approach for using intermittent data such as
satellite-scale observations.

## 5. Conclusions

Empirical pdfs derived from soil saturation observations provided key information to determine unknown ecohydrological parameters $s^*$, $s_w$, $E_{max}$, and $E_w$. This study documented a generalizable Bayesian inversion framework to accurately infer parameters of the stochastic soil water balance model and their associated uncertainty using freely available rainfall and soil
moisture observations at point, footprint and satellite scales. Optimal analytical soil saturation pdfs were consistent with empirical pdfs. Uncertainty in parameter estimates was smallest when the number of unknown parameters was reduced to three, assuming a constant relation between $E_{max}$ and $E_w$ among sites. The proposed framework was found to be robust. Accurate results were obtained using sparse subsets of the datasets, demonstrating the advantage of analyzing soil saturation pdfs instead of time series. The Bayesian framework was also used to evaluate the sensitivity of the soil water balance model
to ecohydrological parameters at varying scales and locations. We demonstrated that the form of the simple ecohydrological model for soil saturation pdfs was in agreement with observations from point, footprint, and satellite scales; however optimal parameters were different at each scale because co-located and concurrent soil saturation pdfs are different at each scale and



may result from spatial heterogeneity in soil water dynamics. Methods developed in this study can be applied in future studies to better understand differences in soil water dynamics at different scales and improve the scaling of ecohydrological processes. Estimates of $s^*$ and $s_w$ were generally not sensitive to the soil depth at which data were measured. Results demonstrated the value of near-surface soil moisture observations to improve the characterization of soil water dynamics at

ecosystem scales.

**Data and code availability**

All datasets used in this study were downloaded from publicly available sources: point-scale soil moisture and rainfall data are available through FLUXNET2015 (http://fluxnet.fluxdata.org/data/fluxnet2015-dataset/); footprint-scale soil moisture data are available through COSMOS (http://cosmos.hwr.arizona.edu/Probes/probelist.html); remotely-sensed soil moisture

data are available through ESA CCI (http://www.esa-soilmoisture-cci.org/node/145); remotely sensed rainfall data are available through NASA TRMM (https://pmm.nasa.gov/data-access/downloads/trmm); global soil texture data are available through FAO HWSD (http://www.fao.org/soils-portal/soil-survey/soil-maps-and-databases/harmonized-world-soil-database-v12/en/). Custom scripts in the Python computing language associated with this analysis are available upon request through a private gitHub repository and will be made publicly available after revisions of this manuscript. (Citation and doi TBD)

**Competing interests**

Authors declare that they have no conflict of interest.

**Acknowledgments**

This material is based upon work supported by the National Science Foundation Graduate Research Fellowship under Grant No. 1314109-DGE. S.P.G. acknowledges the financial support of the Unites States National Aeronautics

and Space Administration (NNX16AN13G). This work used the Extreme Science and Engineering Discovery Environment (XSEDE) via allocation DEB160018, which is supported by National Science Foundation grant number ACI-1548562. This work used data acquired and shared by the FLUXNET community, including these networks: AmeriFlux, AfriFlux, AsiaFlux, CarboAfrica, CarboEuropeIP, CarboItaly, CarboMont, ChinaFlux, Fluxnet-Canada, GreenGrass, ICOS, KoFlux, LBA, NECC, OzFlux-TERN, TCOS-Siberia, and USCCC. The FLUXNET eddy covariance data processing and

harmonization was carried out by the European Fluxes Database Cluster, AmeriFlux Management Project, and Fluxdata project of FLUXNET, with the support of CDIAC and ICOS Ecosystem Thematic Center, and the OzFlux, ChinaFlux and AsiaFlux offices.





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





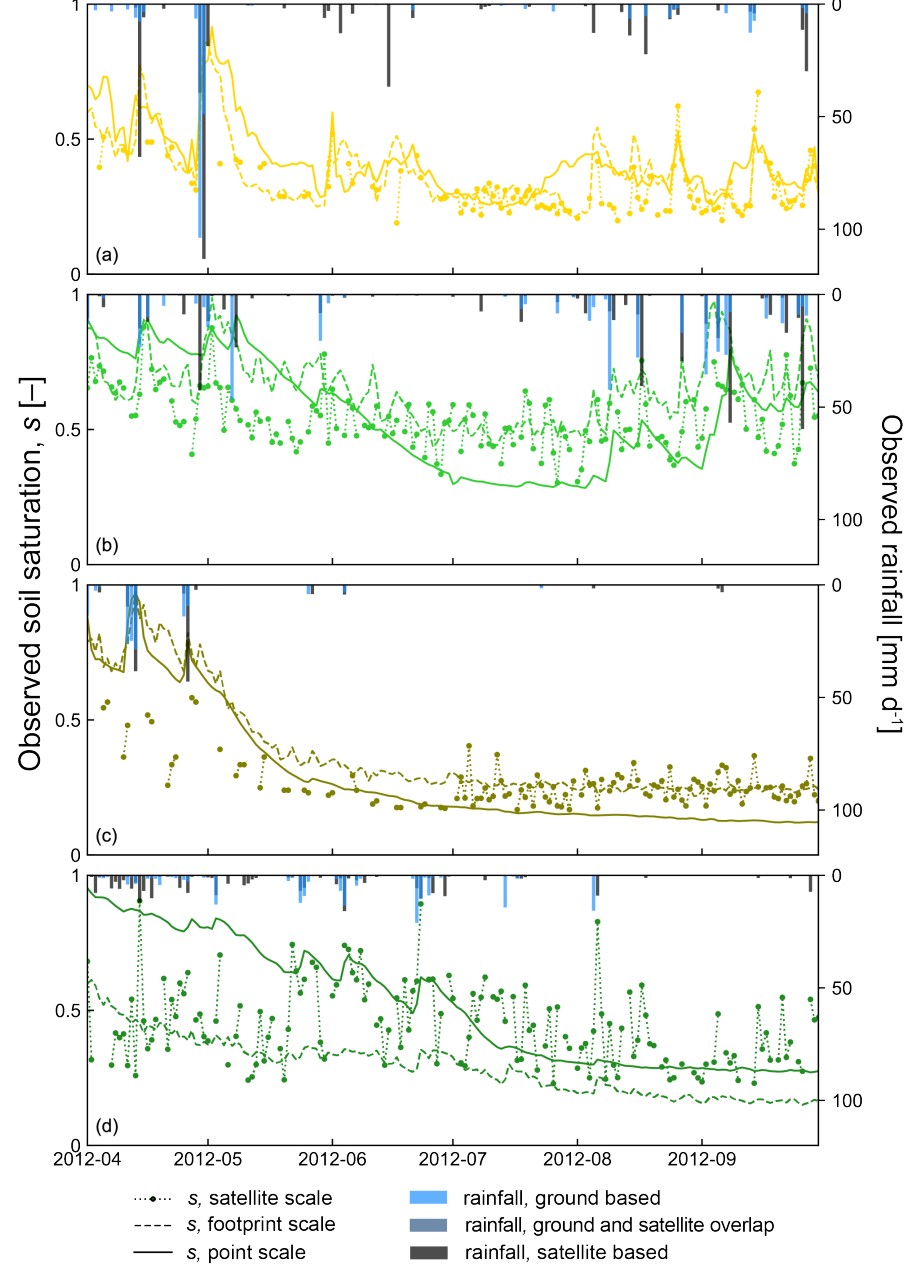

**Figure 1: Soil saturation and rainfall time series from (a) US-ARM, (b) US-MMS, (c) US-Ton, and (d) US-Me2.**





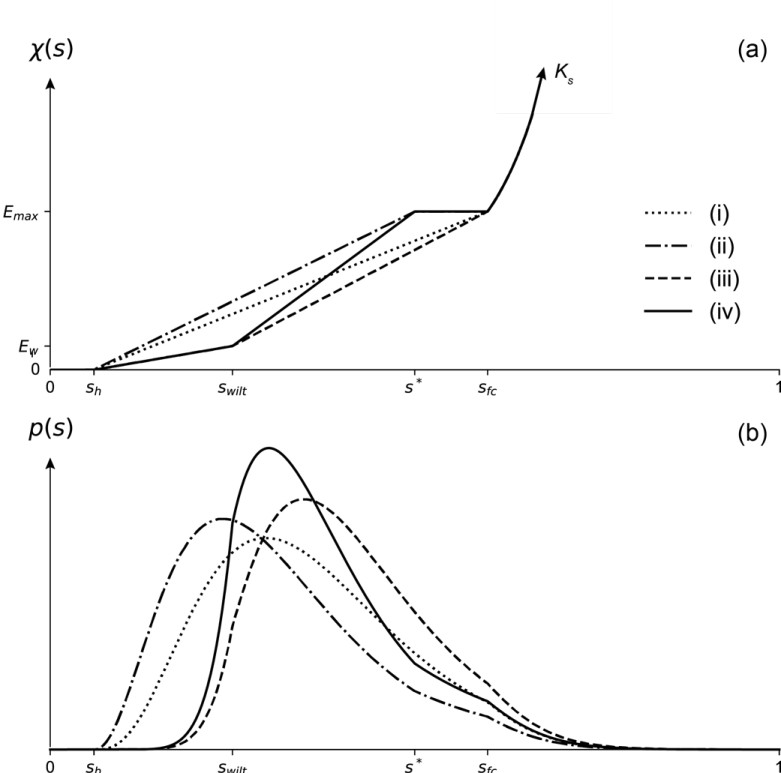

**Figure 2: Conceptual illustration of (a) soil water losses as a function of soil-saturation states, $\chi(s)$, for a loamy soil and (b) associated probability density functions of soil saturation, $p(s)$, for a sub-tropical climate. Increasing levels of model complexity (i – iv) are defined in Table 2.**



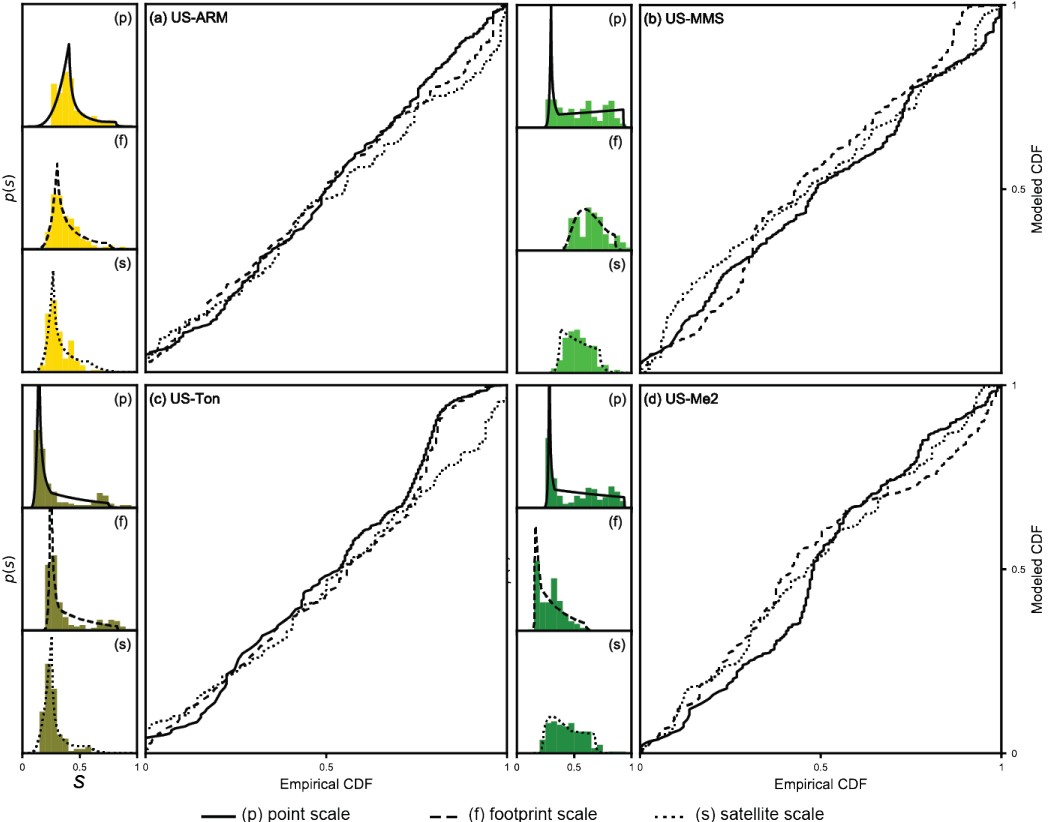

**Figure 3: Empirical verses modelled soil saturation probability distribution (p(s)) and cumulative density functions (CDF) for (a) US-ARM; (b) US-MMS; (c) US-TON; (d) US-Me2; (p) point scale; (f) footprint scale; (s) satellite scale. The mean values of the posteriori parameter distributions were used with model variation (iv') and each spatial scale's sensing depth.**





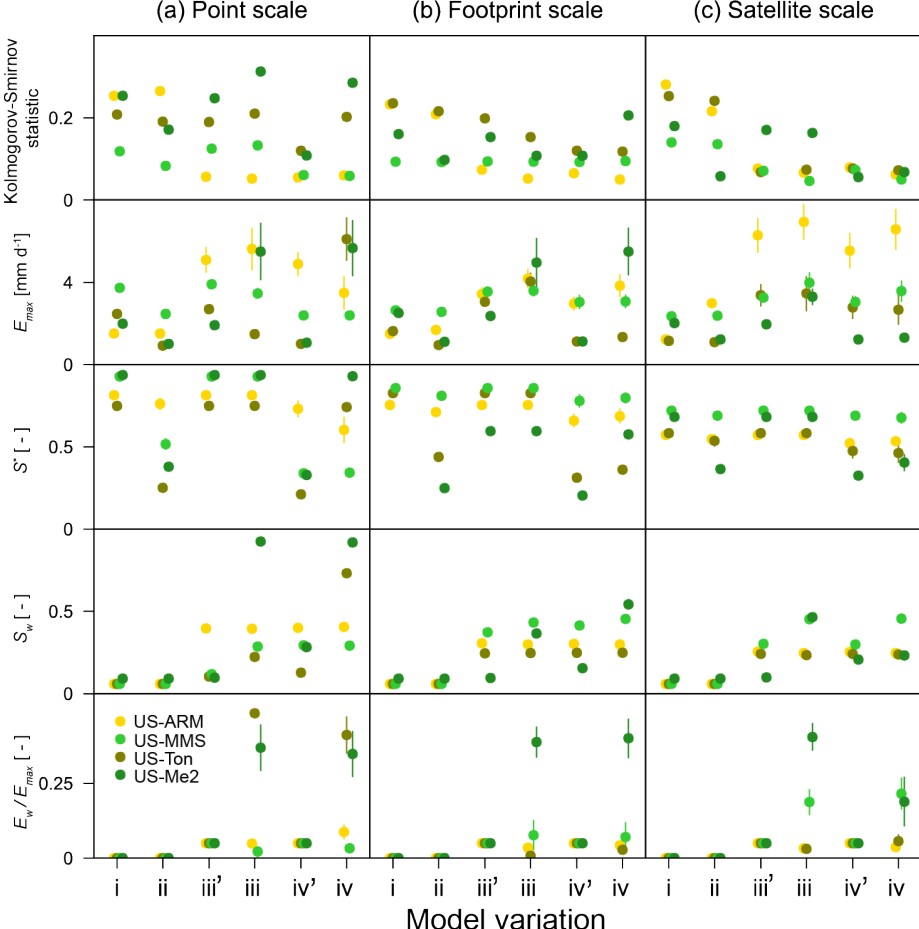

**Figure 4: Goodness of fit and ecohydrological model parameters inferred with increasing model complexity. Model variations i –iv are defined in Table 2; the median results of the converged model inversions are plotted; error bars represent the standard deviations of the posteriori distribution of 50 thousand random parameters samples resulting from the MH-MCMC algorithm.**





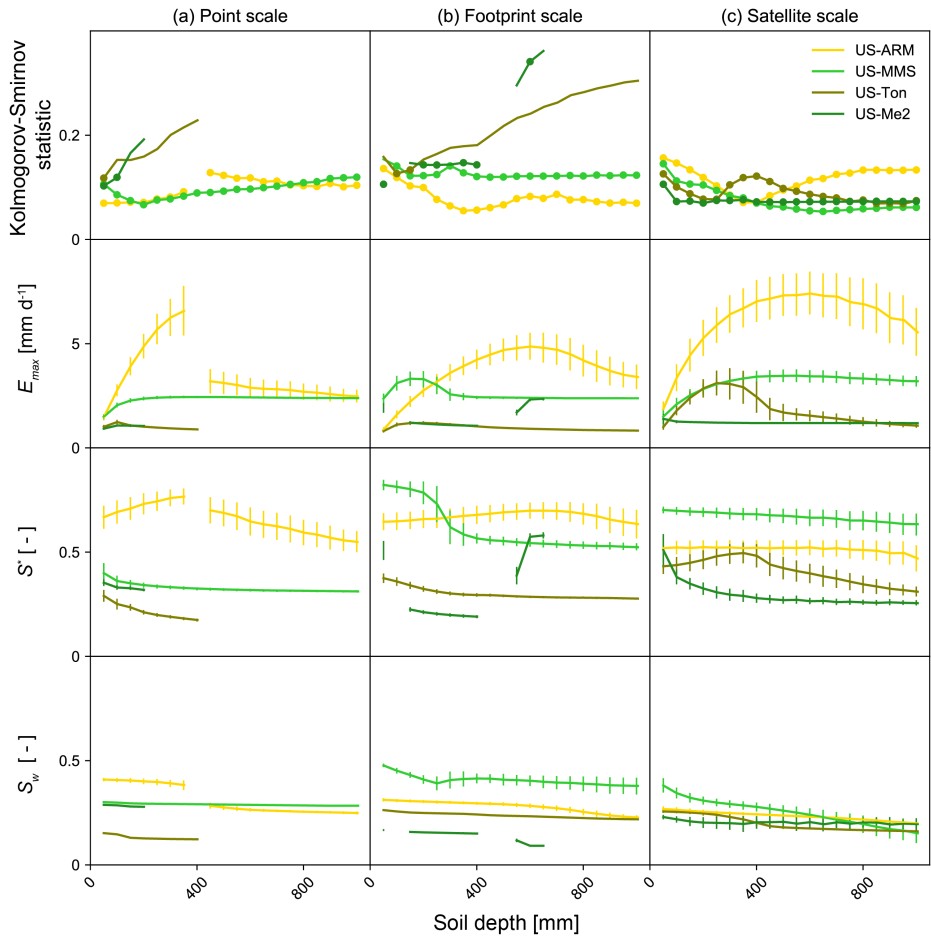

**Figure 5: Goodness of fit and ecohydrological model parameters inferred with soil depths ranging from 5 cm to 1m. The median results of the converged model inversions are plotted; circular markers indicate that the Kolmogorov-Smirnov statistic is significant with a 95 % confidence level; error bars represent the standard deviations of the posteriori distribution of 50 thousand random parameters samples resulting from the MH-MCMC algorithm.**



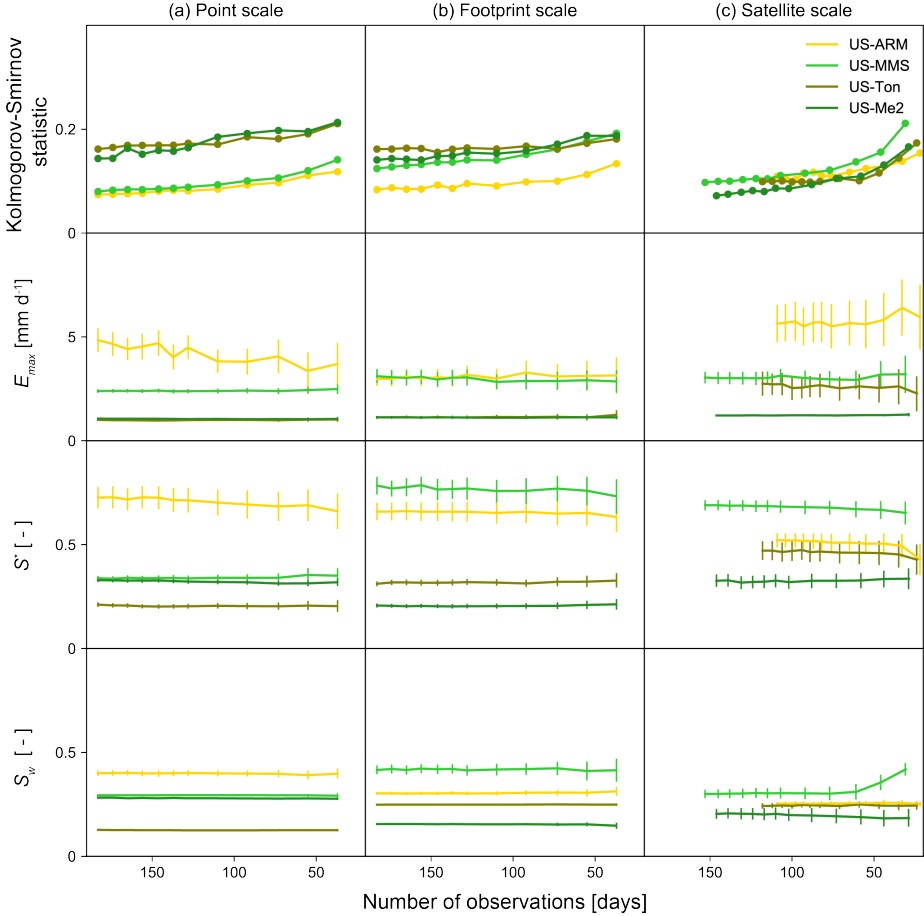

**Figure 6: Goodness of fit and ecohydrological model parameters inferred with decreasing number of soil saturation observations. The median results of the converged model inversions are plotted; circular markers indicate that the Kolmogorov-Smirnov statistic is significant with a 95 % confidence level; error bars represent the standard deviations of the posteriori distribution of 50 thousand random parameters samples resulting from the MH-MCMC algorithm.**





**Table 1 – Selected study sites**

| Site Name | ARM Southern Great Plains | Morgan Monroe State Forest | Tonzi Ranch | Metolius Mature Ponderosa Pine |
|---|---|---|---|---|
| **FLUXNET2015 ID** | US-ARM | US-MMS | US-Ton | US-ME2 |
| **COSMOS ID** | 15 | 27 | 32 | 38 |
| **Latitude** | 36.6058 (36.625) | 39.3232 (39.375) | 38.4316 (38.375) | 44.4523 (44.375) |
| **Longitude** | -97.4888 (-97.375) | -86.4131 (-86.375) | -120.966 (-120.87) | -97.4888 (-97.375) |
| **Elevation [m]** | 314 | 275 | 177 | 1253 |
| **Vegetation** | Crops and grassland | Deciduous forest | Oak savanna | Ponderosa pine forest |
| **MAT [°C]** | 14.8 | 10.9 | 15.8 | 6.3 |
| **MAP [mm]** | 843 | 1032 | 559 | 523 |
| **Soil Texture** | Loam | Loam | Loam | Sandy Loam |
| $n$ [-] | $0.35^{(p)}, 0.34^{(f)}, 0.46^{(s)}$ | $0.46^{(p)}, 0.66^{(f)}, 0.43^{(s)}$ | $0.53^{(p)}, 0.39^{(f)}, 0.43^{(s)}$ | $0.36^{(p)}, 0.59^{(f)}, 0.41^{(s)}$ |
| $K_s$ [mm day$^{-1}$] | 317 | 317 | 317 | 622 |
| $b$ [-] | 4.55 | 4.55 | 4.55 | 3.11 |
| $s_h$ [-] | 0.06 | 0.06 | 0.06 | 0.09 |
| $s_{fc}$ [-] | $0.81^{(p)}, 0.75^{(f)}, 0.57^{(s)}$ | $0.93^{(p)}, 0.86^{(f)}, 0.72^{(s)}$ | $0.94^{(p)}, 0.60^{(f)}, 0.68^{(s)}$ | $0.94^{(p)}, 0.60^{(f)}, 0.68^{(s)}$ |
| $\alpha$ [mm day$^{-1}$] | $26.9^{(p, f)}, 24.5^{(s)}$ | $10.7^{(p, f)}, 13.3^{(s)}$ | $9.7^{(p, f)}, 14.6^{(s)}$ | $4.8^{(p, f)}, 3.0^{(s)}$ |
| $\lambda$ [day$^{-1}$] | $0.05^{(p, f)}, 0.10^{(s)}$ | $0.22^{(p, f)}, 0.20^{(s)}$ | $0.07^{(p, f)}, 0.04^{(s)}$ | $0.20^{(p, f)}, 0.39^{(s)}$ |

Latitude and longitude in parenthesis correspond the centroid of the satellite area associated with the site location; MAT, mean annual temperature from long-term FLUXNET2015 data; MAP, mean annual precipitation from long-term FLUXNET2015 data; Soil texture taken from the HWSD; $n$, porosity; $K_s$, saturated soil hydraulic conductivity; $b$, pore size distribution index; $s_h$, hydroscopic point; $s_{fc}$, field capacity; $\alpha$, observed average daily rainfall depth (April – September, 2012); $\lambda$, observed average daily rainfall frequency (April – September, 2012; superscripts $^{(p)}$, $^{(f)}$, and $^{(s)}$ correspond to values used for the point-, footprint-, and satellite-scale analysis. Citations for each FLUXNET2015 site: Sebastien Biraud (2002–) AmeriFlux US-ARM ARM Southern Great Plains site- Lamont, 10.17190/AMF/1246027; Kim Novick, Rich Phillips (1999–) AmeriFlux US-MMS Morgan Monroe State Forest, 10.17190/AMF/1246080; Bev Law (2002–) AmeriFlux US-Me2 Metolius mature ponderosa pine, 10.17190/AMF/1246076; Dennis Baldocchi (2001–) AmeriFlux US-Ton Tonzi Ranch, 10.17190/AMF/1245971





**Table 2. Model variations**

|        | $i$ | $s_w$ | $s^*$ | $E_{max}$ | $E_w$ |
|--------|-----|-------|-------|-----------|-------|
| (i)    | 1   | $s_h$ | $s_{fc}$ | –      | 0     |
| (ii)   | 2   | $s_h$ | –     | –         | 0     |
| (iii') | 2   | –     | $s_{fc}$ | –      | $0.05E_{max}$ |
| (iii)  | 3   | –     | $s_{fc}$ | –      | –     |
| (iv')  | 3   | –     | –     | –         | $0.05E_{max}$ |
| (iv)   | 4   | –     | –     | –         | –     |

$i$, number of unknown parameters; –, indicates that a parameter is inferred from the model inversion; $s_w$, field capacity; $s^*$, point of incipient stomatal closure; $s_h$, hydroscopic point; $s_{fc}$, field capacity; $E_{max}$, maximum evapotranspiration; $E_w$, evaporation at the wilting point