# Peer review of "Probabilistic inference of ecohydrological parameters using observations from point to satellite scales"

_Hydrology and Earth System Sciences, 2017_

## Referee Comment (RC1) · M. Zhang (Referee) · 15 Dec 2017

I. General comments

Thank you for the opportunity to review the paper "Probabilistic inference of ecohydrological parameters using observations from point to satellite scales". This work introduces a Bayesian inference technique that estimates four ecohydrological parameters from empirical soil moisture pdfs. The paper's novelty lies in the application of this technique beyond the point scale. In the method, the four ecohydrological parameters, which encompass soil water holding thresholds and evapotranspiration, were related to soil moisture observations through Laio et al. (2001)'s analytical formula. The authors

then pose questions about the spatial scale, data availability, and model complexities that are appropriate for such an estimation method, and provide concise answers: estimates are most robust at the satellite scale; the method is accurate with as few as 75 random daily observations; and a specific group of parameters (sw, s*, Emax, Ew = 0.05Emax) can be inferred with highest accuracy. In my opinion, this paper, with major revisions, will have important implications in hydrological modeling. Below are my scientific comments, requests for clarification, and technical corrections.

II. Major comments

1. Applicability of the method

I appreciated the paper's use of sensitivity tests to define the method's applicability in a range of data availability levels, spatial scales, rooting depths, and model complexities. However, I think there's room for another, broader view of method applicability. The conclusions about method applicability were (naturally) only applied in cases where the simulation converges. It would be important to also define the conditions under which the method does (or does not) behave well. On page 1 lines 15-16, the authors wrote that "parameter estimates were most constrained for scales and locations at which soil water dynamics are more sensitive to the fitted ecohydrological parameters of interest". Am I correct in concluding that the method does not converge when soil moisture is NOT sensitive to the ecohydrological parameters of interest?

I recommend that the authors address the conditions under which the method fails to converge. They have briefly mentioned the effect of dry vs. wet climates, but I would like to see a discussion on the effects of soil and vegetation type as well.

2. Choice of estimated parameters

On page 3 line 3, the authors state that the method focuses on estimating "vegetation controls on soil water dynamics". Within this broad category of parameters, four were chosen specifically: sw, s*, Ew, and Emax. The authors should elucidate their choice

of parameters in two ways.

First, there should be a brief explanation of why four was chosen as the maximum number of parameters. If it was out of concern for equifinality, a formal analysis should be included.

Second, I was surprised to see that the rooting depth Z was not among the estimated parameters. From my point of view, Z could be estimated in the same manner as the four chosen parameters and significantly affects the soil moisture pdf. Porporato's work indicates that the volume of storage in the rooting zone is a key determinant of the pdf shape, so there is an a priori reason to expect that Z is an important parameter. In Section 4.2, the authors mentioned that the four estimated parameters aren't very sensitive to the value of Z, but I'm not convinced that Figure 5 supports this conclusion. I strongly suggest a practical or theoretical explanation about why Z was not chosen as an estimated parameter.

III. Minor comments

Section 2.2.1: Model definition

In my opinion, ignoring interception is questionable given the differences in forest type (and especially the presence of deciduous forest in some sites). I recommend a defense of the decision to ignore interception in the soil moisture model.

Using a date range of April to September might introduce nonstationary behavior in climate parameters as the seasons progress from spring to autumn. I suggest a discussion of the impact of (1) nonstationary Emax within this period due to vegetation growth, particularly leaf out and LAI changes in the deciduous forest sites; and (2) any large changes in rainfall occurrence in summer-dry climates on the method's accuracy.

Section 2.2.2: Climate, soil and vegetation parameter characterization

On page 7 lines 17-18, the authors provided a reasonable explanation for why sfc, sh and Ks don't significantly affect soil moisture pdf. It would be nice, though not crucial,

to support this claim using either a sensitivity analysis or with reference to existing analytical studies from Laio et al., (2001).

Section 2.3.1: Application of the Bayes theorem

The authors have assumed uninformed prior knowledge of each of the soil balance parameters while applying Bayes theorem. However, the soil type, climate, and primary forms of vegetation are known at each site, and soil threshold parameters may be estimated from pedotransfer functions. Therefore it seems that an informed prior for each of the four parameters was in fact possible. I suggest exploring the influence of including informed priors on the results and, based on this exploration, defend or reject the decision to use an uninformed prior.

Section 4: Results and Discussion

Several times over the course of this section, the authors mentioned that "acceptable results" were obtained in the various sensitivity tests. The authors should define what is meant by "acceptable" earlier on.

The Kolmogorov-Smirnov statistic is subject to bias and therefore a problematic way to compare pdfs. I recommend exploring measures that compare pdf quantiles, as was done in Muller et al. (2014).

In addition to comparing pdfs, I recommend validating values of the individual estimated parameters. For example, estimations of Emax should be compared to Emax calculated from the Hargreaves equation, and estimates of s* and sw should be compared to results from pedotransfer functions.

Section 4.1: Level of model complexity

Based on Figure 4, it looks like certain location-parameter pairs are very sensitive to model complexity, whereas others are not. I recommend that the authors further explore and explain this sensitivity.

Section 5: Conclusions

I suggest including proposed next steps to improve this method, or planned applications using this method.

Figures

Figure 1: In general, satellite scale soil moisture seems to fluctuate much more than that of footprint scale under dry climate conditions. The caption should include a comment on why this is so, and on the implications of this on performance at the satellite scale.

Figure 4: In the caption, explain why are there error bars associated with only some data points.

Figure 5: In the caption, explain the abrupt changes and "dangling" data points around soil depths of 400m and 600mm for the point and footprint scale plots, respectively.

Figures 4 to 6: please add a legend showing that each of the different colors represents a different location.

IV. Technical corrections

Page 1 line 13: be more specific about what is meant by "footprint" scale.

Page 1 line 25: "back to the atmosphere"

Page 2 line 29: "space-borne"

Page 6 line 9: "commonly used in soil water balance"

Page 9 line 20: the run was discarded"

Page 9 line 21: "more than 10 run samples"

The paper skips directly from section 2 to section 4.

Figure 3 caption: "empirical versus modeled"

Reference

Muller, M.F, D. N. Dralle, and S. E. Thompson (2014), Analytical model for flow duration curves in seasonally dry climates, Water Resour. Res., 50, 5510-5531, doi: 10.1002/2014WR015301.

---

## Referee Comment (RC2) · M. ăF. Muller (Referee) · 20 Dec 2017

The authors use soil moisture observations in a Bayesian inversion procedure to estimate vegetation-related drivers of soil moisture dynamics in the root zone, as modeled by a simple model of soil moisture distribution. The authors apply the approach to a diverse sample of study regions where soil moisture and climate observations are available at different scales. The presented research is important and innovative in that it investigates the potential for recent remote sensing approaches that monitor spatially aggregated soil moisture to estimate eco-hydrologic parameters that are very challenging to observe in-situ, even in well instrumented basins. The research also

bridges the gap between different observation scales, which has potentially interesting implications in poorly gauged regions. While I recommend the paper for publication in HESS, I would also like to raise a few comments/questions that could possibly help the authors during the revision of their paper.

Major comments 1. The authors appear to use the same sample of soil moisture observations to calibrate (via Bayesian Inversion) and validate (KS tests and Fig 3) the approach, which instictively raised red flags on a first read. After reflecting, it became clear (well, to me at least) that the purpose of the exercise was to show that Bayesian inversion can be use to estimate vegetation-related drivers of soil moisture using soil moisture time series, conditionnal on the assumed pdf model being an accurate description of soil moisture dynamics. In that case, the research design would be appropriate because the posterior CV portrays estimation uncertainties and the goodness-of-fit shows that the soil moisture model is, indeed, appropriate. Consequently, the purpose of the goodness-of-fit test appears to be to evaluate the functional form of the pdf, not the estimated parameter values, so it is fine to use the same dataset to calibrate parameters and evaluate outcomes. Please clarify the distinct function of these two metrics as appropriate.

2. I am having issues with the way you use KS tests to evaluate pdf fits. First off, if I am not mistaken, the null hypothesis of a ks test is that the two tested distributions are identical. If so, the p-value could be interpreted as the probability of obtaining a ks-distance at least as large as the one that would be obtained if the two samples were taken from the same distribution. This is loosely equivalent to the probability of falsely rejecting the null. In other words, a p-value of 5% would mean that one has a 95% chance of being right when stating that the two distributions are different, which is quite a low standard when assessing goodness of fit. Significance levels don't tell anything about type II errors, which is what I would think we are ultimately after when evaluating goodness of fits. More importantly, the KS satistic does not follow the kolmogorov distribution (i.e. estimated p-values are wrong) if the same sample of data is used to

calibrate the cdf model and construct the empirical cdf to which it is compared. In my opinion, however, a formal test is not necessary to make your point here (see point 1). The graphs in Fig 3 are sufficient to make the point that the laio model reproduces the shape of the observed empirical histogram. You can then use a distance measure to monitor fits in the sensitivity analyisis. The KS-distance is probably not the most appropriate measure for that though, as it only considers the largest distance between the cdfs — global distance metrics like the Cramer Van Mises statistic or quantile-level nash sutcliffe efficiency, Muller 2016), or information based criteria (e.g, AIC, Ceola 2010) are useful alternatives to consider.

3. Your sensitivity analysis on soil depth (Section 4.2.) convinces me that the value assumed for Z in eqn 2 has little effect on the modeled soil moisture dynamics. This is of course important, but without actually measuring whole column soil moisture, I fail to see how you test the homogeneity assumption (i.e. that near surface soil moisture observations can be used to estimate whole-column characteristics). Please elaborate.

4. I would find it interesting to elaborate on the interpretation of convergence in the context of Bayesian inversion. You mention (I think) that MCMC runs do not converge if insufficient information is available in the empirical p(s) to determine the considered model parameters. I would find it interesting to elaborate on when (and why) these non converging runs arise, perhaps in your discussion on data availability (section 4.3).

5. Finally, I would find it useful for get a sense of how parameters estimated using SM observations taken at a certain scale are valid at different scales. This would have interesting implications, for instance in terms of using satellite remote sensing SM observations to estimate smaller scale SM dynamics in ungauged regions. You discuss this point a little in the paper, but it would be interesting to substantiate your arguments with some analysis. For instance you could run a goodness of fit analysis between modeled SM distributions using params estimated at one scale to empirical SM pdfs observed at another scale.

Minor comments

p3. I would find it useful if you could comment on the advantages of using the Bayesian inversion approach you propose vs more "standard" frequentist approaches such as maximal likelihood, which is the go-to approach I would take to fit a "low dimensional" (4 params) closed form analytical pdf.

p7 l.18. To illustrate your claim, it would be useful if you could present statistics on the frequency of s in each zone of the pdf (in eqn 2) using your best estimation of s* and sw at each site.

p7. Please describe your procedure to compute empirical pdf's from time series observation. If you use kernels to estimate density functions, please specify and justify the chosen shape and bandwith.

p9 l.20: 'discarded'

p10: section 3 is missing

p11. It would be useful to summarize the results (model, scale, posterior CV, goodness of fit distance) for the different cases of the sensitivity analysis in a table.

p12 l.27. "Consistent" has a very specific statistical meaning (asymptotically unbiased), please rephrase if necessary

p13 l 18: "versus"

p 14 l3. Please elaborate on how you could disentangle confounding effects of scale and observation depths. The way I understand it, your analysis in Section 4.2 shows that the results are insensitive to the assumed root-zone depth, not the actual depth, which appears to be unknown (see point 3 above).

Fig 5: you state that the Kolmogorov statistic is significant with a 95% confidence levels. Does that mean that the statistic is significantly different from zero? If so, I would interpret that as having a 5% chance of being wrong if I state that the two compared

[Figure]

distributions are different (see my point on KS tests above), which I don't think is the point you intended to make.

References

Ceola, Serena, et al. "Comparative study of ecohydrological streamflow probability distributions." Water Resources Research 46.9 (2010).

Müller, M. F., and S. E. Thompson. "Comparing statistical and process-based flow duration curve models in ungauged basins and changing rain regimes." Hydrology and Earth System Sciences 20.2 (2016): 669-683.

---

## Referee Comment (RC3) · D. Dralle (Referee) · 21 Dec 2017

REVIEW of the paper, "Probabilistic inference of ecohydrological parameters using observations from point to satellite scales" (Bassiouni et al.)

Submitted for possible publication in HESS

The authors pair in situ and remotely sensed soil moisture data with a Bayesian approach to infer parameters in a 1-d analytical model for soil moisture dynamics.

General Comments:

1) My primary concern is that the authors frequently claim "accurate" results, yet the

study does not include any comparison between predicted and measured soil moisture thresholds. I would say that the study is more accurately described as an exercise in Bayesian model calibration. The novelty of the study, in my opinion, lies in comparing parameters of calibrated PDFs across observation scales. This is a useful exercise, though it's not fully explored in the study. The authors only go so far as to say that "spatial heterogeneity" explains shifting parameter values across scales. The significance of the study would be greatly increased if the authors worked to address some of these scaling effects. A couple questions include: How transferrable are inferred parameter values between scales? How might the optimal form of the PDF change across scales if heterogeneity is the culprit? And, are there simple in silico exercises that could be performed to explore these questions? For example, if the authors generate spatially correlated fields of soil moisture parameters and solve the 1-d model at each point, can aggregation explain (even qualitatively) observed trends in the inferred parameters? What are the implications for applications in sparsely monitored areas, or for making useful predictions at a point using remotely sensed data?

2) While I appreciate the authors' thoroughness, the inclusion of 6 distinct models for soil moisture dynamics somewhat obscures the paper's results. What intuition does this degree of added complexity provide, other than "model performance increases when there are more parameters to tune"? Could some of these results be relegated to Supporting Information, keeping the two most illustrative models?

3) The authors assume steady-state conditions for application of the stochastic models. While this may be appropriate for MMS and ARM, soil moisture dynamics at the seasonally dry sites Tonzi Ranch and Metolius are highly non-stationary during the dry season study months April – September. One can see this in the bi-modality of the soil moisture PDFs in Figure 3. At the very least, it is important for the authors to address or test the effects of this non-stationarity on inferred parameter values. How might strong non-stationarity affect the interpretability of parameter inferences? Perhaps more appropriately, the authors could consider related models that can accommodate seasonally dry soil moisture dynamics. In particular, Dralle et al. (2016, doi: 10.1002/2015wr017813) develop a seasonal stochastic soil moisture model and apply the model at Tonzi Ranch. The calibrated parameter values in that study are exactly comparable to inferred values in the present study. Similarly, Viola et al. (2008, doi: 10.1029/2007WR006371) present a transient formulation of the same stochastic soil moisture model.

Specific Comments:

Lines 8-9: What is a "hydrologically meaningful" scale?

Lines 9-10: Passive voice makes the sentence a little confusing; try, "we hypothesize that pdfs of soil saturation encode sufficient information..."

Line 12: When the authors refer to soil "saturation", do they mean "water content", or "moisture"? I associate the word "saturation" with a water content equal to porosity.

Line 28: Check spelling of reference.

Line 31: What are the "mean components of the soil water balance"?

Line 17: Issues with citations

Line 18: "interference"?

Lines 1-2: Usage, "confront pdfs...to a commonly used analytical model"?

Lines 3-4: I do not believe the model specifies that ET occurs at a constant rate Emax.

Line 12: Do Rawls (1982) list physical soil characteristics for these sites?

Lines 9-10: It's not clear to me why values for Ew/Emax must be tested in a separate (not shown) calibration procedure. See General Comment (2). Page 12

Lines 6-7: My understanding of Emax is that it quantifies atmospheric moisture demand. Why should it scale with rooting depth? Typically, I've seen this value computed using Penman-Monteith e.g. Viola et al. (2008, doi: 10.1029/2007WR006371).

Line 1: I would suggest that model performance at Tonzi and Metolius suffers primarily due to the stationarity assumption, which is likely not valid at these Mediterranean sites.

---

## Referee Comment (RC4) · X. Feng (Referee) · 21 Dec 2017

The manuscript titled "Probabilistic inference of ecohydrological parameters using observations from point to satellite scales" by Bassiouni et al. adopts a Bayesian inference approach to estimate parameters from a parsimonious soil moisture model based on readily available data (soil texture, rainfall, soil moisture) at the point, footprint, and satellite scales. This is a worthwhile exercise and paves the way for the evaluation of the utility of soil moisture data from satellite products. I recommend its publication contingent on clarification on a few issues.

1. A key assumption embedded in the use of this approach requires that the time

series of soil moisture capture the whole range of realizable values. This is required to disentangle cases where soil moisture values cannot be observed due to physical constraints (e.g., imposed by saturation thresholds – the point of this study) versus heuristic constraints (e.g., we simply have not measured it under sufficiently wet or dry conditions). Please include this caveat and discuss practical considerations in overcoming this issue.

2. Relatedly, the study concludes that "model inference at wetter sites . . . is more successful than at dry sites" because known rainfall parameters have been used to constrain the model at wetter sites, where it is hypothesized to play a stronger role in determining the soil moisture pdf. I think this is true, but does not capture the whole story. The "drier" sites used in this study (Tonzi Ranch and Metolius) are also located in Mediterranean climates where substantial seasonal variations in soil moisture can occur between early summer (April/May) and late summer (Sept), which span the period of study. This is apparent from inspection of Figure 1, where soil moisture undergoes an initial rapid decay in Tonzi and Metolius.

As such, I suspect that this assumption of steady state may impact the following statement which I found very interesting (Page 11, line 15): "sw was more important in the analytical equation for soil saturation pdfs and soil water loss equations than s*." If the time series span a transient period that eventually converge toward a dry state, then the shape of the soil moisture pdf would be less defined around s* because there would be relatively fewer soil moisture values near s* than near sw. In that case, sw would naturally become a more important parameter because the shape of the soil moisture pdf would be more defined around sw, but this would be purely an artifact of the relative data availability around sw and s*. To test this issue, I think it might be useful to divide the time series into more distinct periods of "wet," "transition," or "dry" and use those periods to explicitly estimate the relevant parameters sfc, s*, and sw.

And a tangential note on Page 6, line 22 "this framework was derived under the assumption of steady state, wherein parameters are constant for a given period of time."

Constant parameter values are not sufficient criteria for achieving steady state – as it can also result in a transient period based on initial conditions. Please be careful with this terminology.

3. The role of rooting depth. While the model-data fit was not greatly affected by different rooting depths, the resulting values for Emax certainly was. Thus, the authors were able to demonstrate equifinality of results by using Emax to compensate for changes in Z. If the goal is to ultimately estimate meaningful values of vegetation and hydrological thresholds from data, is model-data fit a sufficient metric for evaluation of this approach? My own take away from this part of the study was that rooting depth can in fact be a very sensitive parameter due to the large amount of change in Emax required to achieve similar fit with data. Perhaps a more useful way of tacking this question would be to include Z as another model parameter and evaluate the site and climate conditions under which its impacts would be limited.

4. A few definitions: Page 1, line 14: "parameter uncertainties" – how are these defined? Page 11, line 9: "the most successful parameter estimations were obtained... with 97, 94, 85 percent converging results" – how are these percentages defined (via GR diagnostics?) and what is the significance of the different levels of convergence? I couldn't find a reference in the text. Minor point: section 4 (results and discussion) should actually be section 3.

---

## Author Comment (AC1) · 22 Feb 2018

**"Probabilistic inference of ecohydrological parameters using observations from point to satellite scales" by Maoya Bassiouni et al.**

**Response to Minghui Zhang (Referee #1)**

5    *I. General comments*

*Thank you for the opportunity to review the paper "Probabilistic inference of ecohydrological parameters using observations from point to satellite scales". This work introduces a Bayesian inference technique that estimates four ecohydrological parameters from empirical soil moisture pdfs. The paper's novelty lies in the application of this technique beyond the point scale. In the method, the four ecohydrological parameters, which encompass soil water holding thresholds*
10    *and evapotranspiration, were related to soil moisture observations through Laio et al. (2001)'s analytical formula. The authors then pose questions about the spatial scale, data availability, and model complexities that are appropriate for such an estimation method, and provide concise answers: estimates are most robust at the satellite scale; the method is accurate with as few as 75 random daily observations; and a specific group of parameters (sw, s\*, Emax, Ew = 0.05Emax) can be inferred with highest accuracy. In my opinion, this paper, with major revisions, will have important implications in*
15    *hydrological modeling. Below are my scientific comments, requests for clarification, and technical corrections.*

**Thank you for your thorough review and constructive suggestions. We have provided responses and preliminary corrections below.**

20
*II. Major comments*

*1. Applicability of the method*
*I appreciated the paper's use of sensitivity tests to define the method's applicability in a range of data availability levels,*
25    *spatial scales, rooting depths, and model complexities. However, I think there's room for another, broader view of method applicability. The conclusions about method applicability were (naturally) only applied in cases where the simulation converges. It would be important to also define the conditions under which the method does (or does not) behave well. On page 1 lines 15-16, the authors wrote that "parameter estimates were most constrained for scales and locations at which soil water dynamics are more sensitive to the fitted ecohydrological parameters of interest". Am I correct in concluding that*
30    *the method does not converge when soil moisture is NOT sensitive to the ecohydrological parameters of interest? I recommend that the authors address the conditions under which the method fails to converge. They have briefly mentioned the effect of dry vs. wet climates, but I would like to see a discussion on the effects of soil and vegetation type as well.*

**Thank you for this suggestion. We agree that this is an interesting aspect of this study and will amend the results and discussion section to elaborate on the interpretation of convergences. The convergence and**
35    **inference uncertainty obtained through the Bayesian approach provides insight on (1) whether the data is consistent with the model form used: whether the model is not complex enough or too complex and equifinality arises and (2) whether the assumptions necessary in the model are met by the data: whether the data spans an appropriate range of values or whether the data meets the stationarity assumption. There was no evidence of effects of soil and vegetation type on the accuracy and convergence of the results. We**
40    **generally concluded that the best results are obtained when model assumptions are appropriately met and therefore the empirical pdf is shaped by the parameters defined in the analytical model. We expect that estimated soil saturation thresholds ($s_w$ and $s^*$) will have greater certainty if the empirical soil saturation pdf is most defined around those values and greater uncertainty if there are relatively fewer soil saturations values observed around the thresholds. Thus $s_w$ may be more certain for drier sites and $s^*$ may be more**
45    **certain at wet sites. If the range of observed values is not representative of the soil moisture pdf because it is truncated by missing observations or affected by noise in the data, parameter estimates may have biases.**

*2. Choice of estimated parameters*
50    *On page 3 line 3, the authors state that the method focuses on estimating "vegetation controls on soil water dynamics". Within this broad category of parameters, four were chosen specifically: sw, s\*, Ew, and Emax. The authors should elucidate their choice of parameters in two ways.*
*First, there should be a brief explanation of why four was chosen as the maximum number of parameters. If it was out of concern for equifinality, a formal analysis should be included.*
55    *Second, I was surprised to see that the rooting depth Z was not among the estimated parameters. From my point of view, Z could be estimated in the same manner as the four chosen parameters and significantly affects the soil moisture pdf. Porporato's work indicates that the volume of storage in the rooting zone is a key determinant of the pdf shape, so there is*

*an a priori reason to expect that Z is an important parameter. In Section 4.2, the authors mentioned that the four estimated parameters aren't very sensitive to the value of Z, but I'm not convinced that Figure 5 supports this conclusion. I strongly suggest a practical or theoretical explanation about why Z was not chosen as an estimated parameter.*

**Thank you for this comment. Practical reasons determined the choice of the four parameters that were estimated. Among all the parameters necessary to compute the analytical soil saturation pdf in Equation 2, four ($s_w$, $s^*$, $E_w$, and $E_{max}$) are not directly observable and generally not reliably estimated using available data and existing methods. The other parameters including rainfall characteristics ($\lambda$ and $\alpha$) and physical soil parameters ($s_{fc}$, $s_h$, $K_s$, and b) were characterized based on readily available data and established methods explained in section 2.2.2.**

**Z was not included as a parameter to be estimated because it is most appropriate for Z to be equal to the measurement depth associated with each scale. Our analysis shows that estimates of sw and s\* are not very sensitive to the depth Z assumed in the model inversion (Emax scales as expected with Z). This is important if the sensing depth is not precisely known or is variable in time and space, which is the case for the cosmos and satellite measurements. We have previously tested the model inversion including Z as a parameter to be estimated. We found in this case, a decreased the number of MH-MCMC runs that converge without significantly increasing goodness of fit because there is equifinality between pairs of Z and $E_{max}$.**

**We will remove the sensitivity test related to soil depth because it is not useful to determine whether estimates of $s_w$ and s\* derived from surface soil moisture measurements are relevant to deeper soil depths and does not provide information on the homogeneity assumption. We will amend the methods section to clarify the choice of setting Z to the measurement depth**

*III. Minor comments*

*Section 2.2.1: Model definition*

*In my opinion, ignoring interception is questionable given the differences in forest type (and especially the presence of deciduous forest in some sites). I recommend a defence of the decision to ignore interception in the soil moisture model.*

**We agree that interception is an important component of the soil water balance at forested sites. In this analysis we decided to apply the simplest form of the soil water balance model that would be consistent with the empirical soil saturation pdfs and did not include interception. Results for forested sites were acceptable and did not indicate that the level of model complexity needed to be increased by including interception. The proposed methods can be modified for other studies in which it is important to include interception as a known or unknown parameter (the code associated with this analysis that will be also published included interception as a parameter, here set to 0). Errors due to ignoring interception at the forested sites in this study may have been absorbed in other estimated parameters such as $E_{max}$ or compensated by uncertainties in observed rainfall characteristics. We added the following sentences of section 2.2.1 to clarify this point.**

For simplification, we assume that the rainfall applied is equal to the amount reaching the ground surface and do not account for rainfall intercepted by vegetation. Interception may be a significant component of the soil water balance at forested sites and may need to be accounted for in other studies.

*Using a date range of April to September might introduce nonstationary behavior in climate parameters as the seasons progress from spring to autumn. I suggest a discussion of the impact of (1) nonstationary Emax within this period due to vegetation growth, particularly leaf out and LAI changes in the deciduous forest sites; and (2) any large changes in rainfall occurrence in summer-dry climates on the method's accuracy.*

**We acknowledge that the date range may not be optimal for stationary behaviour in climate parameters at each site. Nevertheless, we decided to select concurrent and consistent time periods for all data sources and sites. Results therefore revealed which sites had poorer goodness of fit statistics and for which the steady-state solution for the analytical soil saturation pdf may not have been most appropriate because of seasonality in the selected period. We will amend the discussion to argue this point. (see responses to other RCs).**

*Section 2.2.2: Climate, soil and vegetation parameter characterization*

*On page 7 lines 17-18, the authors provided a reasonable explanation for why sfc, sh and Ks don't significantly affect soil moisture pdf. It would be nice, though not crucial, to support this claim using either a sensitivity analysis or with reference to existing analytical studies from Laio et al., (2001).*

**Ok this reference will be added here. We also added to Table 1 minimum and maximum observed soil saturation values (April-September, 2012) for comparison with soil saturation threshold estimates.**

*Section 2.3.1: Application of the Bayes theorem*

*The authors have assumed uninformed prior knowledge of each of the soil balance parameters while applying Bayes theorem. However, the soil type, climate, and primary forms of vegetation are known at each site, and soil threshold parameters may be estimated from pedotransfer functions. Therefore, it seems that an informed prior for each of the four parameters was in fact possible. I suggest exploring the influence of including informed priors on the results and, based on this exploration, defend or reject the decision to use an uninformed prior.*

**We acknowledge that some information about the soil type, climate, and primary forms of vegetation are**

known at each site. We have taken advantage of this knowledge in defining the parameters that were not estimated in the model (see Table 1) and defining the bounds of parameters to estimate (Eq 3) therefore better constraiingn the estimated parameters and avoiding equifinality. The added complexity of informed prior knowledge was unecessary. Our goal was to develop a method with the minimum level of complexity in order for it to be applied at any location using easily available data, which is particularly important for the satellite scale analysis. If pedotransfers are not well defined and inconsistent with the soil moisture data they unnecessary uncertainty would introduced in the methods.

*Section 4: Results and Discussion*
*Several times over the course of this section, the authors mentioned that "acceptable results" were obtained in the various sensitivity tests. The authors should define what is meant by "acceptable" earlier on.*

**We will revise section 2.3.2. to explicitly state the evaluation goals and metrics used.**

Optimal analytical soil saturation pdfs are evaluated by the following criteria.

(1) The Bayesian inversion converges and the Gelman-Rubin diagnostic approaches 1 for each estimated parameter (<1.1).
(2) There is goodness of fit between the optimum analytical pdf derived from the mean parameter estimates and the empirical pdfs derived from observations using the Kolmogorov-Smirnov (KS) statistic and the quantile level Nash-Sutcliffe efficiency (NSE) (Müller et al., 2016).
(3) Posterior distributions of parameter estimates are physically plausible and have low coefficients of variations.

**We will also revised the Results section to specify the criteria values that are described and discussed as acceptable.**

*The Kolmogorov-Smirnov statistic is subject to bias and therefore a problematic way to compare pdfs. I recommend exploring measures that compare pdf quantiles, as was done in Muller et al. (2014).*

**This is a good suggestion. We agree that the KS test has disadvantages. We have reported it because was the most strict in quantifying divergence between the analytical and empirical pdfs. In contrast, the NSE values were almost always greater than 0.95 and were less useful. We will report both the KS and NSE in the revision.**

*In addition to comparing pdfs, I recommend validating values of the individual estimated parameters. For example, estimations of Emax should be compared to Emax calculated from the Hargreaves equation, and estimates of s\* and sw should be compared to results from pedotransfer functions.*

**We agree that it would be useful to validate estimated parameters with other estimates. However, estimated ecohydrological parameters that are generally not directly measured. This is also argued in Miller et al., 2007). It is therefore challenging to compare estimated parameter values to site-specific observations and determine their accuracy because these are not directly available. We can relate estimated parameters to calibration efforts of comparable parameters from previous studies.**

**$E_{max}$ is not exactly the atmospheric moisture demand, it is a fraction of the atmospheric moisture demand that can be withdrawn from the soil layer considered. $E_{max}$ can be equal to the atmospheric moisture demand approximated by potential evapotranspiration (PET) if the full soil column or rooting depth is considered. In this study we cannot assume that $E_{max}$ = PET because only the surface soil moisture is sensed. It was not meaningful to compare s\* and sw to estimates from pedotransfer functions because these functions are highly non-linear and not specifically calibrated for data used at each site/scale.**

*Section 4.1: Level of model complexity*
*Based on Figure 4, it looks like certain location-parameter pairs are very sensitive to model complexity, whereas others are not. I recommend that the authors further explore and explain this sensitivity.*

**OK, we will add a comments related to the differences in uncertainty for certain sites in Figure 4.**

Section 5: Conclusions
I suggest including proposed next steps to improve this method, or planned applications using this method.

**OK, we will add the following sentences to the conclusions**

This study provided a method to estimate ecohydrological characteristics that are not directly observable, and for which established estimation methods are not available. This study only used available datasets from sensor networks and global satellite products and methods can therefore be applied to a large range of sites or to full global datasets to improve understanding of spatial patterns in ecohydrological parameters relevant for local and global water cycle analyses.**.**

*Figures*

*Figure 1: In general, satellite scale soil moisture seems to fluctuate much more than that of footprint scale under dry climate conditions. The caption should include a comment on why this is so, and on the implications of this on performance at the satellite scale.*

**Thank you for noticing this. We are not aware of references that analysed causes of higher noise in the satellite-scale soil moisture observations during dry periods. We are not able to make any clear interpretations of this pattern based on the short observation period and limited sites presented in this study. Data indicates that the noise in the satellite-scale soil moisture observations does not significantly affect the mean of the observed soil moisture but may have increased the kurtosis of the empirical pdfs. We will report the mean, standard deviation and kurtosis of empirical soil moisture pdfs in Table 1. Overall, we do not expect our methods to be affected by the noise in the satellite data at the selected locations. This is an illustration of the advantage of analysing pdfs versus time series (mentioned in our introduction) to estimate ecohydrological parameters from satellite soil moisture data. Often areas with highly uncertain satellite soil moisture observations are masked out data products and should not be an issue. Future studies should always assess data quality related to this potential problem**

**We will revise the following sentence in section 2.1 Data Analysed:**

Soil saturation and rainfall data at each scale and for each site during the selected analysis period are presented in Fig. 1 and summary statistics are reported in Table 1. The difference in data quality between data sources and sites is not expected to significantly affect empirical soil saturation pdfs and resulting parameter estimates in this study.

*Figure 4: In the caption, explain why are there error bars associated with only some data points.*

**Error bars are not visible if the standard deviation is smaller than the plot marker, we will add this statement in the legend. Error bars were different for the different figures because not every figure took into account the same soil depths (Z). The revised figures will only show results using Z equal to the sensing depths for each scale. Error bars representing the standard deviations of the estimated parameters will be consistent for all revised figures and different from figures in the previous draft.**

*Figure 5: In the caption, explain the abrupt changes and "dangling" data points around soil depths of 400m and 600mm for the point and footprint scale plots, respectively.*

**Because the revision will remove the sensitivity test related to soil depth, this figure will also be removed.**

*Figures 4 to 6: please add a legend showing that each of the different colors represents a different location.*

**Figures 4 to 6 have a legend with each location's name, we will add the title 'Site name' to the legend to increase clarity.**

*IV. Technical corrections*

*Page 1 line 13: be more specific about what is meant by "footprint" scale.*

**The footprint scale is specifically defined in Section 2.1.**

*Page 1 line 25: "back to the atmosphere"*

**OK this will be corrected**

*Page 2 line 29: "space-borne"*

**OK this will be corrected**

*Page 6 line 9: "commonly used in soil water balance"*

**OK this will be corrected**

*Page 9 line 20: the run was discarded"*

**OK this will be corrected**

*Page 9 line 21: "more than 10 run samples"*

**OK this will be corrected**

*The paper skips directly from section 2 to section 4.*

**OK this will be corrected**

*Figure 3 caption: "empirical versus modelled"*

**OK this will be corrected**

*Reference*
*Muller, M.F, D. N. Dralle, and S. E. Thompson (2014), Analytical model for flow duration curves in seasonally dry climates, Water Resour. Res., 50, 5510-5531, doi: 10.1002/2014WR015301.*

---

## Author Comment (AC2) · 22 Feb 2018

**"Probabilistic inference of ecohydrological parameters using observations from point to satellite scales" by Maoya Bassiouni et al.**

**Response to Marc F. Müller (Referee #2)**

5

*The authors use soil moisture observations in a Bayesian inversion procedure to estimate vegetation-related drivers of soil moisture dynamics in the root zone, as modeled by a simple model of soil moisture distribution. The authors apply the approach to a diverse sample of study regions where soil moisture and climate observations are available at different scales. The presented research is important and innovative in that it investigates the potential for recent remote sensing approaches*
10 *that monitor spatially aggregated soil moisture to estimate eco-hydrologic parameters that are very challenging to observe in-situ, even in well instrumented basins. The research also bridges the gap between different observation scales, which has potentially interesting implications in poorly gauged regions. While I recommend the paper for publication in HESS, I would also like to raise a few comments/questions that could possibly help the authors during the revision of their paper.*

15 **Thank you for your thorough review and constructive suggestions. We have provided responses and some preliminary corrections below.**

*Major comments*

*1. The authors appear to use the same sample of soil moisture obervations to calibrate (via Bayesian Inversion) and validate*
20 *(KS tests and Fig 3) the approach, which instictively raised red flags on a first read. After reflecting, it became clear (well, to me at least) that the purpose of the exercise was to show that Bayesian inversion can be use to estimate vegetation-related drivers of soil moisture using soil moisture time series, conditionnal on the assumed pdf model being an accurate description of soil moisture dynamics. In that case, the research design would be appropriate because the posterior CV portrays estimation uncertainties and the goodness- of-fit shows that the soil moisture model is, indeed, appropriate.*
25 *Consequently, the purpose of the goodness-of-fit test appears to be to evaluate the functional form of the pdf, not the estimated parameter values, so it is fine to use the same dataset to calibrate parameters and evaluate outcomes. Please clarify the distinct function of these two metrics as appropriate.*

**Yes, the above comment describes our intentions. We will revise section 2.3.2. to explicitly state the evaluation goals and metrics used.**
30 Optimal analytical soil saturation pdfs are evaluated by the following criteria.
   (1) The Bayesian inversion converges and the Gelman-Rubin diagnostic approaches 1 for each estimated parameter (<1.1).
   (2) There is goodness of fit between the optimum analytical pdf derived from the mean parameter estimates and the empirical pdfs derived from observations using the Kolmogorov-Smirnov (KS) statistic and the
35 quantile level Nash-Sutcliffe efficiency (NSE) (Müller et al., 2016).
   (3) Posterior distributions of parameter estimates are physically plausible and have low coefficients of variations.

*2. I am having issues with the way you use KS tests to evaluate pdf fits. First off, if I am not mistaken, the null hypothesis of a*
40 *ks test is that the two tested distributions are identical. If so, the p-value could be interpreted as the probability of obtaining a ks-distance at least as large as the one that would be obtained if the two samples were taken from the same distribution. This is loosely equivalent to the probability of falsely rejecting the null. In other words, a p-value of 5% would mean that one has a 95% chance of being right when stating that the two distributions are different, which is quite a low standard when assessing goodness of fit. Significance levels don't tell anything about type II errors, which is what I would think we*
45 *are ultimately after when evaluating goodness of fits. More importantly, the KS satistic does not follow the kolmogorov distribution (i.e. estimated p-values are wrong) if the same sample of data is used to calibrate the cdf model and construct the empirical cdf to which it is compared. In my opinion, however, a formal test is not necessary to make your point here (see point 1). The graphs in Fig 3 are sufficient to make the point that the laio model reproduces the shape of the observed empirical histogram. You can then use a distance measure to monitor fits in the sensitivity analyisis. The KS-distance is*
50 *probably not the most appropriate measure for that though, as it only considers the largest distance between the cdfs âA˘Tˇ global distance metrics like the Cramer Van Mises statistic or quantile-level nash sutcliffe efficiency, Muller 2016), or information based criteria (e.g, AIC, Ceola 2010) are useful alternatives to consider.*

**We agree that the KS test has disadvantages. The KS test was the most strict in quantifying divergence between the analytical and empirical pdfs. In contrast, the NSE values were almost always greater than 0.95**

and were less useful. We agree that the p-value for the KS test is not always meaningful and will remove p-value details in the plots and text. We will report both KS and NSE values in the revision.

*3. Your sensitivity analysis on soil depth (Section 4.2.) convinces me that the value assumed for Z in eqn 2 has little effect on the modeled soil moisture dynamics. This is of course important, but without actually measuring whole column soil moisture, I fail to see how you test the homogeneity assumption (i.e. that near surface soil moisture observations can be used to estimate whole-column characteristics). Please elaborate.*

**We agree that it is difficult to test the homogeneity assumption through the sensitivity tests in this analysis. We have decided to remove the sensitivity analysis related to Z. We will only consider Z equal to the actual measurement depth for each sensor in the revision.**

*4. I would find it interesting to elaborate on the interpretation of convergence in the context of Bayesian inversion. You mention (I think) that MCMC runs do not converge if insufficient information is available in the empirical p(s) to determine the considered model parameters. I would find it interesting to elaborate on when (and why) these non converging runs arise, perhaps in your discussion on data availability (section 4.3).*

**We agree that this is an interesting aspect of this study and will amend the results and discussion section to elaborate on the interpretation of convergences. The convergence and inference uncertainty obtained through the Bayesian approach provides insight on (1) whether the data is consistent with the model form used: whether the model is not complex enough or too complex and equifinality arises and (2) whether the assumptions necessary in the model are met by the data: whether the data spans an appropriate range of values and whether the data meets the stationarity assumption.**

5. Finally, I would find it useful for get a sense of how parameters estimated using SM observations taken at a certain scale are valid at different scales. This would have interesting implications, for instance in terms of using satellite remote sensing SM observations to estimate smaller scale SM dynamics in ungaged regions. You discuss this point a little in the paper, but it would be interesting to substantiate your arguments with some analysis. For instance you could run a goodness of fit analysis between modeled SM distributions using params estimated at one scale to empirical SM pdfs observed at another scale.

**These are interesting questions that may be better answered with a different dataset. We will add a few sentences describing the potential of the proposed methods to address theses questions in the discussion section and relate to recent references on scaling of the stochastic soil water balance model. Our results indicate that the parameters estimated at one scale are not applicable at other scales. One reason is that soil texture constraints ($s_h$ and $s_{fc}$) are different. Another point is that when averaging over larger areas, the effects of a large number of plants (as opposed to one in a point measurement) will change the $s^*$ and $s_w$. Ideally soil water retention parameters would be accurately known and soil saturation thresholds could be converted to more universal values such as soil water potentials and therefore be more transferable for scaling analysis, assuming $E_{max}$ is uniform within the area.**

*Minor comments*

*p3. I would find it useful if you could comment on the advantages of using the Bayesian inversion approach you propose vs more "standard" frequentist approaches such as maximal likelihood, which is the go-to approach I would take to fit a "low dimensional" (4 params) closed form analytical pdf.*

**This comment will be addressed by revising the following sentence in the introduction:**

We selected a Bayesian inversion approach instead of a maximum likelihood approach because it quantifies the inference uncertainty directly and improves upon the work of Miller et al. (2007), which used a least-squares approach to calibrate soil saturation pdfs. In addition, inference uncertainty provided by the Bayesian approach can be used to evaluate the validity assumptions necessary for the model inversion.

*p7 l.18. To illustrate your claim, it would be useful if you could present statistics on the frequency of s in each zone of the pdf (in eqn 2) using your best estimation of s\* and sw at each site.*

**We will visualize s\* and sw in Figure 3 to address this suggestion. Also, we will report minimum and maximum observed soil saturation for each site and scale in Table 1.**

*p7. Please describe your procedure to compute empirical pdf's from time series observation. If you use kernels to estimate density functions, please specify and justify the chosen shape and bandwith.*

**Empirical pdfs were visualized with histograms in Figure 3 using 20 bins, evenly spaced between 0 and 1. In the Bayesian inversion, for each observed soil saturation value, we compute the theoretical probability of that value given a set of model parameters. To compute the quantile level NSE, we compare the quantile score of the observations to the theoretical quantile score from the optimal analytical pdf model. To compute**

the KS we compare the *n* observed saturation values with *n* randomly sampled saturation values from the optimal analytical pdf model.

*p9 l.20: 'discarded'*
**This will be corrected**

*p10: section 3 is missing*
**This will be corrected**

*p11. It would be useful to summarize the results (model, scale, posterior CV, goodness of fit distance) for the different cases of the sensitivity analysis in a table.*
**We will consider reporting the most important summary statistics in a results table if these cannot be clearly reported in the text and are not are already visualized in the figures.**

*p12 l.27. "Consistent" has a very specific statistical meaning (asymptotically unbiased), please rephrase if necessary*
**We will rephrased to:**
when the mean and standard deviation of the randomly selected observations were most representative of the full record and therefore consistent with the rainfall characteristics**.**

*p13 l 18: "versus"*
**This will be corrected**

*p 14 l3. Please elaborate on how you could disentangle confounding effects of scale and observation depths. The way I understand it, your analysis in Section 4.2 shows that the results are insensitive to the assumed root-zone depth, not the actual depth, which appears to be unknown (see point 3 above).*
**Yes, our analysis shows that estimates of $s_w$ and $s^*$ are not very sensitive to the depth assumed in the model inversion. This is important if the sensing depth is not precisely known or is variable in time and space, which is the case for the cosmos and satellite measurements. We will remove the sensitivity test related to soil depth because it is not useful to determine whether estimates of $s_w$ and $s^*$ derived from surface soil moisture measurements are relevant to deeper soil depths. We will explain this choice in the revised methods section.**

*Fig 5: you state that the Kolmogorov statistic is significant with a 95% confidence levels. Does that mean that the statistic is significantly different from zero? If so, I would interpret that as having a 5% chance of being wrong if I state that the two compared distributions are different (see my point on KS tests above), which I don't think is the point you intended to make.*
**Yes, that was the point we intended to make, we have decided to remove the details about the KS significance in the figures as response to your comment above.**

*References*

*Ceola, Serena, et al. "Comparative study of ecohydrological streamflow probability distributions." Water Resources Research 46.9 (2010).*
*Müller, M. F., and S. E. Thompson. "Comparing statistical and process-based flow duration curve models in ungauged basins and changing rain regimes." Hydrology and Earth System Sciences 20.2 (2016): 669-683.*

---

## Author Comment (AC3) · 22 Feb 2018

**"Probabilistic inference of ecohydrological parameters using observations from point to satellite scales" by Maoya Bassiouni et al.**

**Response to David Dralle (Referee #3)**

*The authors pair in situ and remotely sensed soil moisture data with a Bayesian approach to infer parameters in a 1-d analytical model for soil moisture dynamics.*

**Thank you for your thorough review and constructive suggestions. We have provided responses and preliminary corrections below.**

*General Comments:*

*1) My primary concern is that the authors frequently claim "accurate" results, yet the study does not include any comparison between predicted and measured soil moisture thresholds. I would say that the study is more accurately described as an exercise in Bayesian model calibration. The novelty of the study, in my opinion, lies in comparing parameters of calibrated PDFs across observation scales. This is a useful exercise, though it's not fully explored in the study.*

**We agree that this study is primarily an exercise in Bayesian calibration of the commonly used stochastic soil water balance model. We explore whether a Bayesian inversion of the model can provide plausible estimates of ecohydrological parameters that are generally not directly measured. It is therefore challenging to compare estimated parameter values to site-specific observations and determine their accuracy because these are not directly available. We can however, as suggested in your minor comments, relate estimated parameters to calibration efforts of comparable parameters from a few previous studies. We will revise section 2.3.2. to explicitly state the evaluation goals and metrics used.**

Optimal analytical soil saturation pdfs are evaluated by the following criteria.

(1) The Bayesian inversion converges and the Gelman-Rubin diagnostic approaches 1 for each estimated parameter (<1.1).
(2) There is goodness of fit between the optimum analytical pdf derived from the mean parameter estimates and the empirical pdfs derived from observations using the Kolmogorov-Smirnov (KS) statistic and the quantile level Nash-Sutcliffe efficiency (NSE) (Müller et al., 2016).
(3) Posterior distributions of parameter estimates are physically plausible and have low coefficients of variations.

**A range of different sites was selected to develop and demonstrate methods in varying environmental conditions. However, the purpose of this study is not to compare estimated values at these sites. We limit the scope of this paper to presenting the model inversion methods and deriving criteria to obtain meaningful parameter estimates. A comparison of estimated parameters can be the focus of a future study in which a larger number of sites are considered and provide more insights on the variability of these ecohydrologic parameters. We will amend the statement of objectives in the introduction and our conclusions to clarify this scope and propose potential future applications.**

*The authors only go so far as to say that "spatial heterogeneity" explains shifting parameter values across scales. The significance of the study would be greatly increased if the authors worked to address some of these scaling effects. A couple questions include: How transferrable are inferred parameter values between scales? How might the optimal form of the PDF change across scales if heterogeneity is the culprit? And, are there simple in silico exercises that could be performed to explore these questions? For example, if the authors generate spatially correlated fields of soil moisture parameters and solve the 1-d model at each point, can aggregation explain (even qualitatively) observed trends in the inferred parameters? What are the implications for applications in sparsely monitored areas, or for making useful predictions at a point using remotely sensed data?*

**These are interesting questions that would require a different dataset. We will add a few sentences describing the potential of the proposed methods to address these questions in the discussion section and relate to recent references on scaling of the stochastic soil water balance model.**

*2) While I appreciate the authors' thoroughness, the inclusion of 6 distinct models for soil moisture dynamics somewhat obscures the paper's results. What intuition does this degree of added complexity provide, other than "model performance*

*increases when there are more parameters to tune"? Could some of these results be relegated to Supporting Information, keeping the two most illustrative models?*

5 **We agree that this section has some information that may be obscuring the main message. We will keep all model alternative and revise the associated figure to only show the evaluation criteria defined section 2.3.2 (KS, NSE, convergence and parameter coefficient of variation). We will revise the description of these results to highlight the importance of the objective of this sensitivity analysis and clarify their relevance to the overall conclusions.**

**While model performance increases with model complexity, the risk of equifinality is also greater and the number of converging independent runs in the MH-MCMC is reduced when the number of parameters to**
10 **tune is increased. The primary reason to evaluate models of increasing complexity was to detect the maximum number of parameters that can be estimated without the risk of equifinality and which are the minimum parameters that need to be fit to have acceptable goodness of fit between the empirical and analytical pdfs. We found that the 3 parameters $s_w$, $s^*$, and $E_{max}$ were necessary and fixing $E_w$ to a small value (5% of $E_{max}$) equifinality was removed. We will amend the discussion of this point in the revision.**

15

*3) The authors assume steady-state conditions for application of the stochastic models. While this may be appropriate for MMS and ARM, soil moisture dynamics at the seasonally dry sites Tonzi Ranch and Metolius are highly non-stationary during the dry season study months April – September. One can see this in the bi-modality of the soil moisture PDFs in Figure 3. At the very least, it is important for the authors to address or test the effects of this non-stationarity on inferred*
20 *parameter values. How might strong non-stationarity affect the interpretability of parameter inferences? Perhaps more appropriately, the authors could consider related models that can accommodate seasonally dry soil moisture dynamics. In particular, Dralle et al. (2016, doi: 10.1002/2015wr017813) develop a seasonal stochastic soil moisture model and apply the model at Tonzi Ranch. The calibrated parameter values in that study are exactly comparable to inferred values in the present study. Similarly, Viola et al. (2008, doi: 10.1029/2007WR006371) present a transient formulation of the same*
25 *stochastic soil moisture model.*

**We agree that seasonality at the Tonzi and Metolius sites affect our ability to inverse the soil water balance model with the selected data. A steady state period could have been better selected for each individual site. For simplicity/consistency we selected a single concurrent period for all sites and scales. Results therefore revealed which sites had poorer goodness of fit statistics and for which the steady-state solution for the**
30 **analytical soil saturation pdf may not have been most appropriate. We will amend the results and discussion to explain this issue at the Tonzi and Metolius sites and propose practical considerations to address it in future studies.**

**We will amend the study goals, methods and discussion to include an additional sensitivity aimed at addressing the issues associated with the steady-state assumption. We will use data from the 2012 record**
35 **periods and compare the goodness of fit of empirical and analytical pdfs using the full year of observations, and dry and wet periods selected specifically for records at each site/scale.**

*Specific Comments:*

*Page 1 Lines 8-9: What is a "hydrologically meaningful" scale?*

**The first sentence of the abstract will be changed to**
40 Vegetation controls on soil moisture dynamics are generally not directly measured directly and not easy to translate into scale and site-specific ecohydrological parameters for simple soil water balance models.

*Page 1 Lines 9-10: Passive voice makes the sentence a little confusing; try, "we hypothesize that pdfs of soil saturation encode sufficient information..."*
45 **The sentence will be changed to:**
We hypothesize that empirical probability density functions (pdfs) of relative soil moisture or soil saturation encodes sufficient information to determine these ecohydrological parameters, and that these parameters can be estimated through inverse modeling of the commonly used stochastic soil water balance.

50 *Page 1 Line 12: When the authors refer to soil "saturation", do they mean "water content", or "moisture"? I associate the word "saturation" with a water content equal to porosity.*

**We will specify :** relative soil moisture or soil saturation

*Page 1 Line 28: Check spelling of reference.*
55 **The spelling will be fixed**

*Page 1 Line 31: What are the "mean components of the soil water balance"?*

**Sentence will be changed to:**

Given this ecohydrological framework, the probability density function (pdf) of soil moisture and the mean components of the soil water balance (rainfall, runoff, evapotranspiration, and leakage losses) are analytically derived

*Page 2 Line 17: Issues with citations*
**The citation will be fixed**

*Page 3 Line 18: "interference"?*
**Interference will be changed to inference**

*Page 4 Lines 1-2: Usage, "confront pdfs. . .to a commonly used analytical model"?*
**We will think of a better word**

*Page 6 Lines 3-4: I do not believe the model specifies that ET occurs at a constant rate Emax.*
**The word constant will be removed and the sentence changed to:**
The rate of evapotranspiration is assumed to occur at a maximum rate ($E_{max}$), which is independent of the saturation state.

*Page 7 Line 12: Do Rawls (1982) list physical soil characteristics for these sites?*
**We will revise the sentence to:**
Physical soil characteristics for soil textures associated with each site, $s_h$, $K_s$, and b were taken from Rawls et al. (1982) and are listed for each site in Table 1.

*Page 8 Lines 9-10: It's not clear to me why values for Ew/Emax must be tested in a separate (not shown) calibration procedure. See General Comment (2).*
**Seer response to General Comment (2). Our results showed that *Ew/Emax needs to be smaller than 10% for equifinality to be reduced and that the convergence, goodness of fit and posteriori parameter distributions were not sensitive to values between 1 and 10%. So we picked 5%. We will amend the methods section to make this point more explicit. We are not including Supplementary material in with this manuscript. However the code associated with the analysis will be published.***

*Page 12 Lines 6-7: My understanding of Emax is that it quantifies atmospheric moisture demand. Why should it scale with rooting depth? Typically, I've seen this value computed using Penman-Monteith e.g. Viola et al. (2008, doi: 10.1029/2007WR006371).*
**$E_{max}$ is not exactly the atmospheric moisture demand, it is a fraction of the atmospheric moisture demand that can be withdrawn from the soil layer considered. $E_{max}$ can be equal to the atmospheric moisture demand approximated by potential evapotranspiration (PET) if the full soil column or rooting depth is considered.**
**In this study we cannot assume that $E_{max}$ = PET because only the surface soil moisture is sensed. In the revised manuscript we will only consider Z equal to the sensing depth and $E_{max}$ is always expected to be lower than PET. We will clarify definitions in section 2.2.2 with the following sentences**
The soil depth considered corresponded to the measurement sensing depths of 10, 20, and 5 cm for the point, footprint, and satellite scales, respectively. Because the soil depth $Z$ is more shallow than the rooting depth, $E_{max}$ is only a fraction of the atmospheric moisture demand (or potential evapotranspiration) contributed by that soil depth and therefore unknown.

*Page 13 Line 1: I would suggest that model performance at Tonzi and Metolius suffers primarily due to the stationarity assumption, which is likely not valid at these Mediterranean sites.*
**We agree and will revise the discussion to reflect this comment.**

---

## Author Comment (AC4) · 22 Feb 2018

**"Probabilistic inference of ecohydrological parameters using observations from point to satellite scales" by Maoya Bassiouni et al.**

**Response to Xue Feng (Referee #4)**

*The manuscript titled "Probabilistic inference of ecohydrological parameters using observations from point to satellite scales" by Bassiouni et al. adopts a Bayesian inference approach to estimate parameters from a parsimonious soil moisture model based on readily available data (soil texture, rainfall, soil moisture) at the point, footprint, and satellite scales. This is a worthwhile exercise and paves the way for the evaluation of the utility of soil moisture data from satellite products. I recommend its publication contingent on clarification on a few issues.*

**Thank you for your thorough review and constructive suggestions. We have provided responses and preliminary corrections below.**

*1. A key assumption embedded in the use of this approach requires that the time series of soil moisture capture the whole range of realizable values. This is required to disentangle cases where soil moisture values cannot be observed due to physical constraints (e.g., imposed by saturation thresholds – the point of this study) versus heuristic constraints (e.g., we simply have not measured it under sufficiently wet or dry conditions). Please include this caveat and discuss practical considerations in overcoming this issue.*

**We agree. This is an important assumption that should be described more explicitly and addressed in the discussion.**

**We will revise the Introduction and add a sentence before presenting the study hypothesis.**

We assume that if a sufficient range of soil moisture values are observed at a site, then the shape of the empirical soil saturation pdf, derived from these observations is constrained by the ecohydrological factors driving soil moisture dynamics. We hypothesize that key information required to determine these ecohydrological factors is encoded in empirical soil saturation pdfs, and that this information can be extracted by calculating the inverse of the commonly used stochastic soil water balance.

**We will also rephrase the introduction of the key questions addressed in this paper.**

What is the minimum amount of data necessary to determine empirical soil saturation pdfs, which are complete and robust enough for estimating ecohydrological parameters through a Bayesian inversion?

**In the Methods, this assumption will be mentioned in section 2.2.2 "Climate, soil and vegetation parameter characterization"**

A key assumption in this analysis is that the whole range of realizable soil moistures values is captured by the selected time series and the empirical soil moisture pdf determined from these observations is not truncated by missing data. In these conditions, the shape of the empirical soil saturation pdf is constrained by the ecohydrological factors driving soil moisture dynamics and the parameters of the analytical pdf can be determined with certainty. In addition, the predetermined value for $s_h$, based on soil texture, is lower than the observed soil saturation observation. This extends the parameter space for $s_w$ in areas that may not be observed in dry enough conditions. It is possible to estimate a value for $s_w$ that is between the minimum observed soil saturation value and $s_h$ and not be observed in the time series. The minimum and maximum observed soil saturation values during the April to September 2012 period are reported in Table 1 to indicate the range of observed soil saturation values used to estimate ecohydrological parameters. We expect that estimated soil saturation thresholds ($s_w$ and $s^*$) will have greater certainty if the empirical soil saturation pdf is most defined around those values and greater uncertainty if there are relatively fewer soil saturations values observed around the thresholds. Thus $s_w$ may be more certain for drier sites and $s^*$ may be more certain at wet sites. If the range of observed values is not representative of the soil moisture pdf because it is truncated by missing observations or affected by noise in the data, parameter estimates may have biases.

**We will also amend the results and discussion section to better evaluate the validity of assumption and relate it to the uncertainty of the parameter estimates and the model inversion convergences. In particular, results from the sub-sampling sensitivity test will be used to discuss this point and describe practical considerations to overcome this issue in future analyses.**

*2. Relatedly, the study concludes that "model inference at wetter sites... is more successful than at dry sites" because known rainfall parameters have been used to constrain the model at wetter sites, where it is hypothesized to play a stronger role in determining the soil moisture pdf. I think this is true, but does not capture the whole story. The "drier" sites used in this study (Tonzi Ranch and Metolius) are also located in Mediterranean climates where substantial seasonal variations in soil*

*moisture can occur between early summer (April/May) and late summer (Sept), which span the period of study. This is apparent from inspection of Figure 1, where soil moisture undergoes an initial rapid decay in Tonzi and Metolius.*

*As such, I suspect that this assumption of steady state may impact the following statement which I found very interesting (Page 11, line 15): "sw was more important in the analytical equation for soil saturation pdfs and soil water loss equations than s*." If the time series span a transient period that eventually converge toward a dry state, then the shape of the soil moisture pdf would be less defined around s* because there would be relatively fewer soil moisture values near s* than near sw. In that case, sw would naturally become a more important parameter because the shape of the soil moisture pdf would be more defined around sw, but this would be purely an artifact of the relative data availability around sw and s*. To test this issue, I think it might be useful to divide the time series into more distinct periods of "wet," "transition," or "dry" and use those periods to explicitly estimate the relevant parameters sfc, s*, and sw.*

**We agree with your interpretation and will amend the results and discussion sections to explain that the assumption of steady state impacts the ability to fit the ecohydrological parameters to data taken during a transitional-dry period at the Tonzi and Metolius sites. We acknowledge that the date range selected may not be optimal for stationary behaviour at each site. A steady state period could have been better selected for each individual site. For simplicity/consistency we selected a single concurrent period for all sites and scales. Results therefore revealed which sites had poorer goodness of fit statistics and for which the steady-state solution for the analytical soil saturation pdf may not have been most appropriate.**

**We will amend the study goals, methods and discussion to include an additional sensitivity aimed at addressing the issues associated with the steady-state assumption. We will use data from the 2012 record periods and compare the goodness of fit of empirical and analytical pdfs using the full year of observations, and dry and wet periods selected specifically for records at each site/scale.**

*And a tangential note on Page 6, line 22 "this framework was derived under the assumption of steady state, wherein parameters are constant for a given period of time."*

*Constant parameter values are not sufficient criteria for achieving steady state – as it can also result in a transient period based on initial conditions. Please be careful with this terminology.*

**We now clarify that the theoretical pdf equation is the steady state solution of the stochastic soil water balance but it does not necessarily imply that the data indicates a steady state.**

*3. The role of rooting depth. While the model-data fit was not greatly affected by different rooting depths, the resulting values for Emax certainly was. Thus, the authors were able to demonstrate equifinality of results by using Emax to compensate for changes in Z. If the goal is to ultimately estimate meaningful values of vegetation and hydrological thresholds from data, is model-data fit a sufficient metric for evaluation of this approach? My own take away from this part of the study was that rooting depth can in fact be a very sensitive parameter due to the large amount of change in Emax required to achieve similar fit with data. Perhaps a more useful way of tacking this question would be to include Z as another model parameter and evaluate the site and climate conditions under which its impacts would be limited.*

**Z was not included as a parameter to be estimated because it is most appropriate for Z to be equal to the measurement depth associated with each measurement. Our analysis shows that estimates of sw and s* are not very sensitive to the depth Z assumed in the model inversion while E$_{max}$ scales as expected with Z. This is important if the sensing depth is not precisely known or is variable in time and space, which is the case for the cosmos and satellite measurements. The model inversion convergence, and the coefficient of variation of posteriori parameter estimates are more important metrics to detect equifinality than goodness of fit. We have previously tested the model inversion including Z as a parameter to be estimated. We found in this case, a decreased the number of MH-MCMC runs that converge without significantly increasing goodness of fit because there is equifinality between pairs of Z and E$_{max}$.**

**In the revised manuscript we will only consider Z equal to the sensing depth. We will remove the sensitivity teste related to soil depth because it is not useful to determine whether estimates of sw and s* derived from surface soil moisture measurements are relevant to deeper soil depths. In the results comparing model complexity alternatives, the figure will be revised to visualize model inversion convergence, and the coefficient of variation of posteriori parameter estimates in addition to goodness of fit. This will clarify the arguments for the choice of models and the explanation related to equifinality.**

*4. A few definitions:*

*Page 1, line 14: "parameter uncertainties" – how are these defined?*

**The term "parameter uncertainties" will be changed. The sentence will read:** the coefficient of variation of posteriori parameter distributions were on average under 10 %.

*Page 11, line 9: "the most successful parameter estimations were obtained. . . with 97, 94, 85 percent converging results" – how are these percentages defined (via GR diagnostics?) and what is the significance of the different levels of convergence? I couldn't find a reference in the text.*

5

We defined in the methods section that the GR diagnostic determines that the algorithm reaches convergence when the within-run variability ($\sigma_w$) is roughly equal to the between-run variability ($\sigma_b$), i.e. $\sigma_w/\sigma_b$ approaches 1. We repeated the optimizations 10 time (with 5 run samples each) for each sensitivity test and the percentage of converging results indicates the percentage of the repeats that had a GR diagnostics for each parameters inferior to 1.1. We considered that a model inversion had appropriately converged if the GR diagnostics was lower than 1.1 for each estimated parameter.

*Minor point: section 4 (results and discussion) should actually be section 3.*
**The numbering will be corrected.**

---

## Author Response (AR2)

**"Probabilistic inference of ecohydrological parameters using observations from point to satellite scales" by Maoya Bassiouni et al.**

**Response to Sally Thompson (Editor)**

*This manuscript received four independent reviews, and all of them offered sensible and constructive suggestions. The reviews broadly concur that the manuscript is of interest to the research community, but that a number of methodological and scope issues need to be addressed. I am largely convinced that these points have been taken on board by your author team based on the response to the reviewers - thank you.*
*To summarize, I will be looking for the following changes in the revised manuscript:*

*(1) Clarify the scope of the paper and avoid making overly strong statements about what it achieves. The goals of model calibration and "identifiability" of parameters (it may be best to speak of identifiability rather than speaking of "avoiding equifinality" -- maybe see Wagener and Kollat 2007?) seem more appropriate than strong statements about correctness and accuracy.*

*(2) Clarify the assumptions in the modelling and how they are addressed. In particular assumptions about statistical stationarity versus parameter stationarity versus homogeneity should be considered.*

*(3) Improve the treatment of non-stationarity in the soil water timeseries. As several reviewers noted, the model was inappropriately inverted during a period of time where soil moisture is non-stationary - drawing conclusions about calibration performance in this situation is of highly questionable value. I reiterate the 3rd reviewers suggestion to consider adopting the frameworks of Dralle (which is simpler) or Viola (more complex) in order to address growing season conditions in Mediterranean sites.*

*(4) Improve and justify the goodness of fit metrics used*

*(5) Tread carefully around the treatment of soil depth versus Emax in the modeling.*

**We appreciate your feedback on our manuscript. We have responded to all of the reviewers suggestions below and addressed your major expectations in the revised draft.**

**(1) We have simplified the scope of the manuscript, revisited the questions addressed in the analysis in response to the reviewers' major comments. A portion of the analysis in the previous draft was no longer relevant and removed from the revised manuscript. We have adopted the word 'identifiability' to avoid making overly strong statements about the parameter estimates.**

**(2) We have reworded the manuscript to clarify the concepts of stationarity and homogeneity.**

**(3) We have used the suggested framework in Dralle and Thompson, 2016 in the revised manuscript to improve treatment of non-stationarity in the soil water time series. Results of the model inversions were improved by the use of a full year time series instead of an isolated summer season. We discuss whether the added complexity of a seasonal model versus a steady state model improves the identifiability of the ecohydrological parameters.**

**(4) We have adopted the quantile-level Nash Sutcliffe Efficiency as an improved goodness of fit metric in the revised manuscript and added section 2.4 and 2.5 to define our evaluation criteria**

**(5) We only consider the soil moisture sensing depth, removed the sensitivity test related to soil depth in the revised manuscript and carefully defined Emax.**

**Response to Minghui Zhang (Referee #1)**

*I. General comments*

*Thank you for the opportunity to review the paper "Probabilistic inference of ecohydrological parameters using observations from point to satellite scales". This work introduces a Bayesian inference technique that estimates four ecohydrological*
5 *parameters from empirical soil moisture pdfs. The paper's novelty lies in the application of this technique beyond the point scale. In the method, the four ecohydrological parameters, which encompass soil water holding thresholds and evapotranspiration, were related to soil moisture observations through Laio et al. (2001)'s analytical formula. The authors then pose questions about the spatial scale, data availability, and model complexities that are appropriate for such an estimation method, and provide concise answers: estimates are most robust at the satellite scale; the method is accurate with as few as 75*
10 *random daily observations; and a specific group of parameters (sw, s\*, Emax, Ew = 0.05Emax) can be inferred with highest accuracy. In my opinion, this paper, with major revisions, will have important implications in hydrological modeling. Below are my scientific comments, requests for clarification, and technical corrections.*

**Thank you for your thorough review and constructive suggestions. We have provided responses and corrections**
15 **below.**

*II. Major comments*

20 *1. Applicability of the method*
*I appreciated the paper's use of sensitivity tests to define the method's applicability in a range of data availability levels, spatial scales, rooting depths, and model complexities. However, I think there's room for another, broader view of method applicability. The conclusions about method applicability were (naturally) only applied in cases where the simulation converges. It would be important to also define the conditions under which the method does (or does not) behave well. On page 1 lines 15-16, the*
25 *authors wrote that "parameter estimates were most constrained for scales and locations at which soil water dynamics are more sensitive to the fitted ecohydrological parameters of interest". Am I correct in concluding that the method does not converge when soil moisture is NOT sensitive to the ecohydrological parameters of interest? I recommend that the authors address the conditions under which the method fails to converge. They have briefly mentioned the effect of dry vs. wet climates, but I would like to see a discussion on the effects of soil and vegetation type as well.*
30 **Thank you for this suggestion. We agree that this is an interesting aspect of this study and have revised the results to report the number of sample runs required to obtain converging results (see method in section 2.3.2). This measure quantifies how rapidly converging results are obtained. Only results that met the convergence criteria were reported. We generally concluded that convergence was obtained when model assumptions are appropriately met and the empirical pdf is consistent with the parameters defined in the analytical model. The revised manuscript**
35 **compared annual pdfs instead of summer season pdfs. This approach was overall more appropriate and model inversions for all sites and datasets converged. There was no evidence of effects of soil and vegetation type on the convergence of the results.**

*2. Choice of estimated parameters*
40 *On page 3 line 3, the authors state that the method focuses on estimating "vegetation controls on soil water dynamics". Within this broad category of parameters, four were chosen specifically: sw, s\*, Ew, and Emax. The authors should elucidate their choice of parameters in two ways.*
*First, there should be a brief explanation of why four was chosen as the maximum number of parameters. If it was out of concern for equifinality, a formal analysis should be included.*
45 *Second, I was surprised to see that the rooting depth Z was not among the estimated parameters. From my point of view, Z could be estimated in the same manner as the four chosen parameters and significantly affects the soil moisture pdf. Porporato's work indicates that the volume of storage in the rooting zone is a key determinant of the pdf shape, so there is an a priori reason to expect that Z is an important parameter. In Section 4.2, the authors mentioned that the four estimated parameters aren't very sensitive to the value of Z, but I'm not convinced that Figure 5 supports this conclusion. I strongly suggest a practical or*
50 *theoretical explanation about why Z was not chosen as an estimated parameter.*
**Thank you for this comment. Practical reasons determined the choice of the parameters that were estimated. Among all the parameters necessary to compute the analytical soil saturation pdf in Equation 2, $s_w$, $s^*$, and $E_{max}$ are not directly observable and generally difficult to estimate using available data and existing methods. The other parameters including rainfall characteristics ($\lambda$ and $\alpha$) and physical soil parameters ($s_{fc}$, $s_h$, $K_s$, and b)**
55 **were characterized based on readily available data and established methods explained in section 2.2.2. We have added a discussion of this choice in the 3rd paragraph of the introduction.**

**Z was not included as a parameter to be estimated because it is most appropriate for Z to be equal to the measurement depth associated with each scale. We have removed the sensitivity test related to soil depth because it is not useful to determine whether estimates of $s_w$ and s* derived from surface soil moisture measurements are relevant to deeper soil depths and does not provide information on the homogeneity assumption. We have defined the choice of setting Z to the measurement depth in the methods section (2.2.2).**

*III. Minor comments*

*Section 2.2.1: Model definition*

*In my opinion, ignoring interception is questionable given the differences in forest type (and especially the presence of deciduous forest in some sites). I recommend a defence of the decision to ignore interception in the soil moisture model.*

**We agree that interception is an important component of the soil water balance at forested sites. In this analysis we decided to apply the simplest form of the soil water balance model that would be consistent with the empirical soil saturation pdfs and did not include interception. Results for forested sites were acceptable and did not indicate that the level of model complexity needed to be increased by including interception. The proposed methods can be modified for other studies in which it is important to include interception as a known or unknown parameter (the code associated with this analysis that will be also published included interception as a parameter, here set to 0). Errors due to ignoring interception at the forested sites in this study may have been absorbed in other estimated parameters such as $E_{max}$ or compensated by uncertainties in observed rainfall characteristics. We added the following sentences of section 2.2.1 to clarify this point.**

For simplification, we assume that the rainfall applied is equal to the amount reaching the ground surface and do not account for rainfall intercepted by vegetation. Interception may be a significant component of the soil water balance at forested sites and may need to be accounted for in other studies.

*Using a date range of April to September might introduce nonstationary behavior in climate parameters as the seasons progress from spring to autumn. I suggest a discussion of the impact of (1) nonstationary Emax within this period due to vegetation growth, particularly leaf out and LAI changes in the deciduous forest sites; and (2) any large changes in rainfall occurrence in summer-dry climates on the method's accuracy.*

**We acknowledge that the date range may not be optimal for stationary behaviour in climate parameters at each site. The revised analysis utilized a full year timeseries and also adopts the framework in Dralle and Thompson (2016) to account for non-stationary dynamics.**

*Section 2.2.2: Climate, soil and vegetation parameter characterization*
*On page 7 lines 17-18, the authors provided a reasonable explanation for why sfc, sh and Ks don't significantly affect soil moisture pdf. It would be nice, though not crucial, to support this claim using either a sensitivity analysis or with reference to existing analytical studies from Laio et al., (2001).*

**OK this reference will be added. We also added to Table 1 minimum and maximum observed soil saturation values for comparison with soil saturation threshold estimates.**

*Section 2.3.1: Application of the Bayes theorem*
*The authors have assumed uninformed prior knowledge of each of the soil balance parameters while applying Bayes theorem. However, the soil type, climate, and primary forms of vegetation are known at each site, and soil threshold parameters may be estimated from pedotransfer functions. Therefore, it seems that an informed prior for each of the four parameters was in fact possible. I suggest exploring the influence of including informed priors on the results and, based on this exploration, defend or reject the decision to use an uninformed prior.*

**We acknowledge that some information about the soil type, climate, and primary forms of vegetation are known at each site. We have taken advantage of this knowledge in defining the parameters that were not estimated in the model (see Table 1) and defining the bounds of parameters to estimate. This was useful to better constrain the estimated parameters and avoiding equifinality. The added complexity of informed prior knowledge was unnecessary. Our goal was to develop a method with the minimum level of complexity in order for it to be applied at any location using easily available data, which is particularly important for the satellite scale analysis. If pedotransfers are not well defined and inconsistent with the soil moisture data they unnecessary uncertainty would introduced in the methods.**

*Section 4: Results and Discussion*
*Several times over the course of this section, the authors mentioned that "acceptable results" were obtained in the various sensitivity tests. The authors should define what is meant by "acceptable" earlier on.*

**Section 2.4 and 2.5 was added in the revised manuscript to explicitly describe evaluation goals and metrics used.**

*The Kolmogorov-Smirnov statistic is subject to bias and therefore a problematic way to compare pdfs. I recommend exploring*

*measures that compare pdf quantiles, as was done in Muller et al. (2014).*

**This is a good suggestion. We agree that the KS test has disadvantages. We reported both the KS and NSE in the revision.**

5 *In addition to comparing pdfs, I recommend validating values of the individual estimated parameters. For example, estimations of Emax should be compared to Emax calculated from the Hargreaves equation, and estimates of s\* and sw should be compared to results from pedotransfer functions.*

**We agree that it would be useful to validate estimated parameters with other estimates. However, estimated ecohydrological parameters that are generally not directly measured. This is also argued in Miller et al., 2007).**

10 **It is therefore challenging to compare estimated parameter values to site-specific observations and determine their accuracy because these are not directly available. Calibration parameters from previous studies are now cited in the revised manuscript. We have used the wording parameter identifiability instead of accuracy to avoid misleading statements.**

**$E_{max}$ is not exactly the atmospheric moisture demand, it is a fraction of the atmospheric moisture demand that can**

15 **be withdrawn from the soil layer considered. $E_{max}$ can be equal to the atmospheric moisture demand approximated by potential evapotranspiration (PET) if the full soil column or rooting depth is considered.**

**In this study we cannot assume that $E_{max}$ = PET because only the surface soil moisture is sensed. It was not meaningful to compare s\* and sw to estimates from pedotransfer functions because these functions are highly non-linear and not specifically calibrated for data used at each site/scale.**

*Section 4.1: Level of model complexity*
*Based on Figure 4, it looks like certain location-parameter pairs are very sensitive to model complexity, whereas others are not. I recommend that the authors further explore and explain this sensitivity.*

**Figure 4 was removed**

Section 5: Conclusions
I suggest including proposed next steps to improve this method, or planned applications using this method.

**OK, we added the following sentences to the conclusions**

This study provided a method to estimate ecohydrological characteristics that are not directly observable, and for which

30 established estimation methods are not available. This study only used available datasets from sensor networks and global satellite products and methods can therefore be applied to a large range of sites or to full global datasets to improve understanding of spatial patterns in ecohydrological parameters relevant for local and global water cycle analyses.

*Figures*

*Figure 1: In general, satellite scale soil moisture seems to fluctuate much more than that of footprint scale under dry climate conditions. The caption should include a comment on why this is so, and on the implications of this on performance at the satellite scale.*

**Thank you for noticing this. We are not aware of references that analysed causes of higher noise in the satellite-**

40 **scale soil moisture observations during dry periods. We are not able to make any clear interpretations of this pattern based on the short observation period and limited sites presented in this study. Data indicates that the noise in the satellite-scale soil moisture observations does not significantly affect the mean of the observed soil moisture but may have increased the kurtosis of the empirical pdfs. We will report the mean, standard deviation and kurtosis of empirical soil moisture pdfs in Table 1. Overall, we do not expect our methods to be affected by**

45 **the noise in the satellite data at the selected locations. This is an illustration of the advantage of analysing pdfs versus time series (mentioned in our introduction) to estimate ecohydrological parameters from satellite soil moisture data. Often areas with highly uncertain satellite soil moisture observations are masked out data products and should not be an issue. Future studies should always assess data quality related to this potential problem**

50 **We revised the following sentence in section 2.1:**

Soil saturation and rainfall data at each scale and for each site during the selected analysis period are presented in Fig. 1 and summary statistics are reported in Table 1. The difference in data quality between data sources and sites is not expected to significantly affect empirical soil saturation pdfs and resulting parameter estimates in this study.

55 *Figure 4: In the caption, explain why are there error bars associated with only some data points.*

**Figure 4 was removed**

*Figure 5: In the caption, explain the abrupt changes and "dangling" data points around soil depths of 400m and 600mm for the point and footprint scale plots, respectively.*

60 **Figure 5 was removed**

*Figures 4 to 6: please add a legend showing that each of the different colors represents a different location.*
**We have changed the location of the legend to increase clarity.**

5 *IV. Technical corrections*
*Page 1 line 13: be more specific about what is meant by "footprint" scale.*
**The footprint scale is specifically defined in Section 2.1.**

*Page 1 line 25: "back to the atmosphere"*
10 **OK this was corrected**

*Page 2 line 29: "space-borne"*
**OK this was corrected**

15 *Page 6 line 9: "commonly used in soil water balance"*
**OK was corrected**

*Page 9 line 20: the run was discarded"*
**OK was corrected**
20
*Page 9 line 21: "more than 10 run samples"*
**OK was corrected**

*The paper skips directly from section 2 to section 4.*
25 **OK was corrected**

*Figure 3 caption: "empirical versus modelled"*
**OK was corrected**

30 *Reference*
*Muller, M.F, D. N. Dralle, and S. E. Thompson (2014), Analytical model for flow duration curves in seasonally dry climates, Water Resour. Res., 50, 5510-5531, doi: 10.1002/2014WR015301.*

**Response to Marc F. Müller (Referee #2)**

*The authors use soil moisture observations in a Bayesian inversion procedure to estimate vegetation-related drivers of soil moisture dynamics in the root zone, as modeled by a simple model of soil moisture distribution. The authors apply the approach to a diverse sample of study regions where soil moisture and climate observations are available at different scales. The presented research is important and innovative in that it investigates the potential for recent remote sensing approaches that monitor spatially aggregated soil moisture to estimate eco-hydrologic parameters that are very challenging to observe in-situ, even in well instrumented basins. The research also bridges the gap between different observation scales, which has potentially interesting implications in poorly gauged regions. While I recommend the paper for publication in HESS, I would also like to raise a few comments/questions that could possibly help the authors during the revision of their paper.*

**Thank you for your thorough review and constructive suggestions. We have provided responses and some corrections below.**

*Major comments*

*1. The authors appear to use the same sample of soil moisture obervations to calibrate (via Bayesian Inversion) and validate (KS tests and Fig 3) the approach, which instictively raised red flags on a first read. After reflecting, it became clear (well, to me at least) that the purpose of the exercise was to show that Bayesian inversion can be use to estimate vegetation-related drivers of soil moisture using soil moisture time series, conditionnal on the assumed pdf model being an accurate de- scription of soil moisture dynamics. In that case, the research design would be appropriate because the posterior CV portrays estimation uncertainties and the goodness- of-fit shows that the soil moisture model is, indeed, appropriate. Consequently, the purpose of the goodness-of-fit test appears to be to evaluate the functional form of the pdf, not the estimated parameter values, so it is fine to use the same dataset to calibrate parameters and evaluate outcomes. Please clarify the distinct function of these two metrics as appropriate.*

**Yes, the above comment describes our intentions. Section 2.4 and 2.5 was added in the revised manuscript to explicitly describe evaluation goals and metrics used.**

*2. I am having issues with the way you use KS tests to evaluate pdf fits. First off, if I am not mistaken, the null hypothesis of a ks test is that the two tested distributions are identical. If so, the p-value could be interpreted as the probability of obtaining a ks-distance at least as large as the one that would be obtained if the two samples were taken from the same distribution. This is loosely equivalent to the probability of falsely rejecting the null. In other words, a p-value of 5% would mean that one has a 95% chance of being right when stating that the two distributions are different, which is quite a low standard when assessing goodness of fit. Significance levels don't tell anything about type II errors, which is what I would think we are ultimately after when evaluating goodness of fits. More importantly, the KS satistic does not follow the kolmogorov distribution (i.e. estimated p-values are wrong) if the same sample of data is used to calibrate the cdf model and construct the empirical cdf to which it is compared. In my opinion, however, a formal test is not necessary to make your point here (see point 1). The graphs in Fig 3 are sufficient to make the point that the laio model reproduces the shape of the observed empirical histogram. You can then use a distance measure to monitor fits in the sensitivity analyisis. The KS-distance is probably not the most appropriate measure for that though, as it only considers the largest distance between the cdfs âA ̆T ̆ global distance metrics like the Cramer Van Mises statistic or quantile-level nash sutcliffe efficiency, Muller 2016), or information based criteria (e.g, AIC, Ceola 2010) are useful alternatives to consider.*

**We reported both KS and NSE values in the revision. We agree that the KS test has disadvantages. We agree that the p-value for the KS test is not always meaningful is not discussed in the revision.**

*3. Your sensitivity analysis on soil depth (Section 4.2.) convinces me that the value assumed for Z in eqn 2 has little effect on the modeled soil moisture dynamics. This is of course important, but without actually measuring whole column soil moisture, I fail to see how you test the homogeneity assumption (i.e. that near surface soil moisture observations can be used to estimate whole-column characteristics). Please elaborate.*

**We agree that it is difficult to test the homogeneity assumption through the sensitivity tests in this analysis. We have removed the sensitivity analysis related to Z. We will only consider Z equal to the actual measurement depth for each sensor in the revision.**

*4. I would find it interesting to elaborate on the interpretation of convergence in the context of Bayesian inversion. You mention (I think) that MCMC runs do not converge if insufficient information is available in the empirical p(s) to determine the considered model parameters. I would find it interesting to elaborate on when (and why) these non converging runs arise, perhaps in your discussion on data availability (section 4.3).*

**We agree that this is an interesting aspect of this study and will amend the results and discussion section to elaborate on the interpretation of convergences. We have revised the results to report (see table 2) the number of sample runs required to obtain converging results (see section 2.3.2). Only results that met the convergence criteria were reported. We generally concluded that convergence was obtained when model assumptions are appropriately met and the empirical pdf is consistent with the parameters defined in the analytical model. The revised manuscript compared annual pdfs instead of summer season pdfs. This approach was overall more appropriate and model inversions for all sites and datasets converged.**

5. Finally, I would find it useful for get a sense of how parameters estimated using SM observations taken at a certain scale are valid at different scales. This would have interesting implications, for instance in terms of using satellite remote sensing SM observations to estimate smaller scale SM dynamics in ungauged regions. You discuss this point a little in the paper, but it would be interesting to substantiate your arguments with some analysis. For instance you could run a goodness of fit analysis between modeled SM distributions using params estimated at one scale to empirical SM pdfs observed at another scale.

**These are interesting questions that may be better answered with a different dataset. Our results indicate that the parameters estimated at one scale are not applicable at other scales. One reason is that soil texture constraints ($s_h$ and $s_{fc}$) are different. Another point is that when averaging over larger areas, the effects of a large number of plants (as opposed to one in a point measurement) will change the $s^*$ and $s_w$. Ideally soil water retention parameters would be accurately known and soil saturation thresholds could be converted to more universal values such as soil water potentials and therefore be more transferable for scaling analysis, assuming $E_{max}$ is uniform within the area.**

*Minor comments*

*p3. I would find it useful if you could comment on the advantages of using the Bayesian inversion approach you propose vs more "standard" frequentist approaches such as maximal likelihood, which is the go-to approach I would take to fit a "low dimensional" (4 params) closed form analytical pdf.*

**This comment is addressed by revising the following sentence in the introduction:**

We selected a Bayesian inversion approach instead of a least-squares or maximum likelihood approach because it quantifies the inference uncertainty directly and improves upon the work of Miller et al. (2007), which used a least-squares approach to calibrate soil saturation pdfs. In addition, measures of inference uncertainty and parameter convergence diagnostics provided by the Bayesian approach can be used to evaluate the validity of model inversion and develop criteria to generalize the presented framework.

*p7 l.18. To illustrate your claim, it would be useful if you could present statistics on the frequency of s in each zone of the pdf (in eqn 2) using your best estimation of s\* and sw at each site.*

**We visualized s\* and sw in Figures 2-5 to address this suggestion. Also, we reported minimum and maximum observed soil saturation for each site and scale in Table 1.**

*p7. Please describe your procedure to compute empirical pdf's from time series observation. If you use kernels to estimate density functions, please specify and justify the chosen shape and bandwith.*

**Empirical pdfs were visualized with histograms in Figure 3 using 20 bins, evenly spaced between 0 and 1. In the Bayesian inversion, for each observed soil saturation value, we compute the theoretical probability of that value given a set of model parameters. The quantile level NSE evaluates quantiles from 1/365 to 354/365.**

*p9 l.20: 'discarded'*

**This was corrected**

*p10: section 3 is missing*

**This was corrected**

*p11. It would be useful to summarize the results (model, scale, posterior CV, goodness of fit distance) for the different cases of the sensitivity analysis in a table.*

**Table 2 reports the most important summary statistics and results.**

*p12 l.27. "Consistent" has a very specific statistical meaning (asymptotically unbiased), please rephrase if necessary*

**We rephrased to:**

because the mean and standard deviation of the randomly selected subsets of annual data were generally representative of the full record

*p13 l 18: "versus"*

**This was corrected**

5     *p 14 l3. Please elaborate on how you could disentangle confounding effects of scale and observation depths. The way I understand it, your analysis in Section 4.2 shows that the results are insensitive to the assumed root-zone depth, not the actual depth, which appears to be unknown (see point 3 above).*

**Yes, our analysis shows that estimates of $s_w$ and $s^*$ are not very sensitive to the depth assumed in the model inversion. This is important if the sensing depth is not precisely known or is variable in time and space, which is**

10    **the case for the cosmos and satellite measurements. We removed the sensitivity test related to soil depth because it is not useful to determine whether estimates of $s_w$ and $s^*$ derived from surface soil moisture measurements are relevant to deeper soil depths. We have defined the choice of setting Z to the measurement depth in the methods section (2.2.2).**

*Fig 5: you state that the Kolmogorov statistic is significant with a 95% confidence levels. Does that mean that the statistic is significantly different from zero? If so, I would interpret that as having a 5% chance of being wrong if I state that the two compared distributions are different (see my point on KS tests above), which I don't think is the point you intended to make.*

**Yes, that was the point we intended to make, we have removed the details about the KS significance in the figures**

20    **as response to your comment above.**

**We agree that this study is primarily an exercise in Bayesian calibration of the commonly used stochastic soil water balance model. We explore whether a Bayesian inversion of the model can provide plausible estimates of ecohydrological parameters that are generally not directly measured. It is therefore challenging to compare estimated parameter values to site-specific observations and determine their accuracy because these are not directly available. We have adopted the wording 'identifiability' to avoid making overly strong statements about the parameter estimates. As suggested in your minor comments, we have cited calibrated parameters from previous studies in te revised manuscript.**
**Section 2.4 and 2.5 was added in the revised manuscript to explicitly describe evaluation goals and metrics used.**

**A range of different sites was selected to develop and demonstrate methods in varying environmental conditions. However, the purpose of this study is not to compare estimated values at these sites. We limit the scope of this paper to presenting the model inversion methods and deriving criteria to obtain meaningful parameter estimates. A comparison of estimated parameters can be the focus of a future study in which a larger number of sites are considered and provide more insights on the variability of these ecohydrologic parameters. We amended the statement of objectives in the introduction and our conclusions to clarify this scope.**

*The authors only go so far as to say that "spatial heterogeneity" explains shifting parameter values across scales. The significance of the study would be greatly increased if the authors worked to address some of these scaling effects. A couple questions include: How transferrable are inferred parameter values between scales? How might the optimal form of the PDF change across scales if heterogeneity is the culprit? And, are there simple in silico exercises that could be performed to explore these questions? For example, if the authors generate spatially correlated fields of soil moisture parameters and solve the 1-d model at each point, can aggregation explain (even qualitatively) observed trends in the inferred parameters? What are the implications for applications in sparsely monitored areas, or for making useful predictions at a point using remotely sensed data?*

**These are interesting questions that would require a different dataset.**

*2) While I appreciate the authors' thoroughness, the inclusion of 6 distinct models for soil moisture dynamics somewhat obscures the paper's results. What intuition does this degree of added complexity provide, other than "model performance increases when there are more parameters to tune"? Could some of these results be relegated to Supporting Information, keeping the two most illustrative models?*

**We agree that this section has some information that may be obscuring the main message. We have removed this analysis from the revised manuscript and only present the comparison of the steady state versus seasonal models with unknown parameters sw, s*, and emax.**

*3) The authors assume steady-state conditions for application of the stochastic models. While this may be appropriate for MMS and ARM, soil moisture dynamics at the seasonally dry sites Tonzi Ranch and Metolius are highly non-stationary during the dry season study months April – September. One can see this in the bi-modality of the soil moisture PDFs in Figure 3. At the very least, it is important for the authors to address or test the effects of this non-stationarity on inferred parameter values. How might strong non-stationarity affect the interpretability of parameter inferences? Perhaps more appropriately, the authors could consider related models that can accommodate seasonally dry soil moisture dynamics. In particular, Dralle et al. (2016, doi: 10.1002/2015wr017813) develop a seasonal stochastic soil moisture model and apply the model at Tonzi Ranch. The calibrated*

*parameter values in that study are exactly comparable to inferred values in the present study. Similarly, Viola et al. (2008, doi: 10.1029/2007WR006371) present a transient formulation of the same stochastic soil moisture model.*

**We agree that seasonality at the Tonzi and Metolius sites affect our ability to inverse the soil water balance model with the selected data. The revised analysis utilized a full year timeseries and also adopts the suggested framework in Dralle and Thompson (2016) to account for non-stationary dynamics.**

*Specific Comments:*

*Page 1 Lines 8-9: What is a "hydrologically meaningful" scale?*

**The first sentence of the abstract was changed to**

Vegetation controls on soil moisture dynamics are generally not directly measured directly and not easy to translate into
10  scale and site-specific ecohydrological parameters for simple soil water balance models.

*Page 1 Lines 9-10: Passive voice makes the sentence a little confusing; try, "we hypothesize that pdfs of soil saturation encode sufficient information. . ."*

**The sentence was changed to:**
15  We hypothesize that empirical probability density functions (pdfs) of relative soil moisture or soil saturation encodes sufficient information to determine these ecohydrological parameters, and that these parameters can be estimated through inverse modelling of the commonly used stochastic soil water balance.

*Page 1 Line 12: When the authors refer to soil "saturation", do they mean "water content", or "moisture"? I associate the word*
20  *"saturation" with a water content equal to porosity.*

**We specify :** relative soil moisture or soil saturation

*Page 1 Line 28: Check spelling of reference.*

**The spelling was fixed**

*Page 1 Line 31: What are the "mean components of the soil water balance"?*

**Sentence was changed to:**

Given this ecohydrological framework, the probability density function (pdf) of soil moisture and the mean components of the soil water balance (rainfall, runoff, evapotranspiration, and leakage losses) are analytically derived

*Page 2  Line 17: Issues with citations*

**The citation was fixed**

*Page 3  Line 18: "interference"?*
35  **Interference was changed to inference**

*Page 4  Lines 1-2: Usage, "confront pdfs. . .to a commonly used analytical model"?*

**We reworded to match pdfs**

40  *Page 6 Lines 3-4: I do not believe the model specifies that ET occurs at a constant rate Emax.*

**The word constant was removed and the sentence was changed to:**

The rate of evapotranspiration is assumed to occur at a maximum rate ($E_{max}$), which is independent of the saturation state.

*Page 7 Line 12: Do Rawls (1982) list physical soil characteristics for these sites?*
45  **The sentence now reads:**

Physical soil characteristics for soil textures associated with each site, $s_h$, $K_s$, and b were taken from Rawls et al. (1982) and are listed for each site in Table 1.

*Page 8 Lines 9-10: It's not clear to me why values for Ew/Emax must be tested in a separate (not shown) calibration procedure.*
50  *See General Comment (2).*

**Seer response to General Comment (2). Our results showed that Ew/Emax needs to be smaller than 10% for equifinality to be reduced and that the convergence, goodness of fit and posteriori parameter distributions were not sensitive to values between 1 and 10%. So we picked 5%. We are not including Supplementary material in with this manuscript. However the code associated with the analysis will be published.**

55

*Page 12 Lines 6-7: My understanding of Emax is that it quantifies atmospheric moisture demand. Why should it scale with rooting depth? Typically, I've seen this value computed using Penman-Monteith e.g. Viola et al. (2008, doi: 10.1029/2007WR006371).*

**$E_{max}$ is not exactly the atmospheric moisture demand, it is a fraction of the atmospheric moisture demand that can be withdrawn from the soil layer considered. $E_{max}$ can be equal to the atmospheric moisture demand approximated**

5 **by potential evapotranspiration (PET) if the full soil column or rooting depth is considered.**

**In this study we cannot assume that $E_{max}$ = PET because only the surface soil moisture is sensed. In the revised manuscript we will only consider Z equal to the sensing depth and $E_{max}$ is always expected to be lower than PET. We clarified definitions in section 2.2.2 with the following sentences**

The soil depth considered corresponded to the measurement sensing depths of 10, 20, and 5 cm for the point, footprint,

10 and satellite scales, respectively. Because the soil depth $Z$ is more shallow than the rooting depth, $E_{max}$ is only a fraction of the atmospheric moisture demand (or potential evapotranspiration) contributed by that soil depth and therefore unknown.

*Page 13 Line 1: I would suggest that model performance at Tonzi and Metolius suffers primarily due to the stationarity assumption,*

15 *which is likely not valid at these Mediterranean sites.*

**We agree, and revised the analysis to discussion this comment.**

**Response to Xue Feng (Referee #4)**

*The manuscript titled "Probabilistic inference of ecohydrological parameters using observations from point to satellite scales" by Bassiouni et al. adopts a Bayesian inference approach to estimate parameters from a parsimonious soil moisture model based on readily available data (soil texture, rainfall, soil moisture) at the point, footprint, and satellite scales. This is a worthwhile exercise and paves the way for the evaluation of the utility of soil moisture data from satellite products. I recommend its publication contingent on clarification on a few issues.*

**Thank you for your thorough review and constructive suggestions. We have provided responses and corrections below.**

*1. A key assumption embedded in the use of this approach requires that the time series of soil moisture capture the whole range of realizable values. This is required to disentangle cases where soil moisture values cannot be observed due to physical constraints (e.g., imposed by saturation thresholds – the point of this study) versus heuristic constraints (e.g., we simply have not measured it under sufficiently wet or dry conditions). Please include this caveat and discuss practical considerations in overcoming this issue.*

**We agree. This is an important assumption that should be described more explicitly and addressed in the revision. A number of sections of the introduction and methods section (2.2.2, 2.4 and 2.5 ) have been revised to consider this suggestion.**

*2. Relatedly, the study concludes that "model inference at wetter sites... is more successful than at dry sites" because known rainfall parameters have been used to constrain the model at wetter sites, where it is hypothesized to play a stronger role in determining the soil moisture pdf. I think this is true, but does not capture the whole story. The "drier" sites used in this study (Tonzi Ranch and Metolius) are also located in Mediterranean climates where substantial seasonal variations in soil moisture can occur between early summer (April/May) and late summer (Sept), which span the period of study. This is apparent from inspection of Figure 1, where soil moisture undergoes an initial rapid decay in Tonzi and Metolius.*
*As such, I suspect that this assumption of steady state may impact the following statement which I found very interesting (Page 11, line 15): "sw was more important in the analytical equation for soil saturation pdfs and soil water loss equations than s*." If the time series span a transient period that eventually converge toward a dry state, then the shape of the soil moisture pdf would be less defined around s* because there would be relatively fewer soil moisture values near s* than near sw. In that case, sw would naturally become a more important parameter because the shape of the soil moisture pdf would be more defined around sw, but this would be purely an artifact of the relative data availability around sw and s*. To test this issue, I think it might be useful to divide the time series into more distinct periods of "wet," "transition," or "dry" and use those periods to explicitly estimate the relevant parameters sfc, s*, and sw.*

**We agree with your comment. The revised analysis utilized a full year timeseries and also adopts the suggested framework in Dralle and Thompson (2016) to account for non-stationary dynamics.**

*And a tangential note on Page 6, line 22 "this framework was derived under the assumption of steady state, wherein parameters are constant for a given period of time."*
*Constant parameter values are not sufficient criteria for achieving steady state – as it can also result in a transient period based on initial conditions. Please be careful with this terminology.*

**We clarified that the theoretical pdf equation is the steady state solution of the stochastic soil water balance but it does not necessarily imply that the data indicates a steady state.**

*3. The role of rooting depth. While the model-data fit was not greatly affected by different rooting depths, the resulting values for Emax certainly was. Thus, the authors were able to demonstrate equifinality of results by using Emax to compensate for changes in Z. If the goal is to ultimately estimate meaningful values of vegetation and hydrological thresholds from data, is model-data fit a sufficient metric for evaluation of this approach? My own take away from this part of the study was that rooting depth can in fact be a very sensitive parameter due to the large amount of change in Emax required to achieve similar fit with data. Perhaps a more useful way of tacking this question would be to include Z as another model parameter and evaluate the site and climate conditions under which its impacts would be limited.*

**Z was not included as a parameter to be estimated because it is most appropriate for Z to be equal to the measurement depth associated with each measurement. Our analysis shows that estimates of sw and s* are not very sensitive to the depth Z assumed in the model inversion while $E_{max}$ scales as expected with Z. This is important if the sensing depth is not precisely known or is variable in time and space, which is the case for the**

**cosmos and satellite measurements. The model inversion convergence, and the coefficient of variation of posteriori parameter estimates were the more important metrics to detect equifinality than goodness of fit. We have previously tested the model inversion including Z as a parameter to be estimated. We found in this case, a decreased in convergergence and no increase in goodness of fit because there is equifinality between pairs of Z and $E_{max}$.**

**In the revised manuscript we only consider Z equal to the sensing depth. We removed the sensitivity teste related to soil depth because it is not useful to determine whether estimates of sw and s\* derived from surface soil moisture measurements are relevant to deeper soil depths.**

*4. A few definitions:*

*Page 1, line 14: "parameter uncertainties" – how are these defined?*

**The term "parameter uncertainties" was changed. The sentence reads:** the coefficient of variation of posteriori parameter distributions were …

*Page 11, line 9: "the most successful parameter estimations were obtained. . . with 97, 94, 85 percent converging results" – how are these percentages defined (via GR diagnostics?) and what is the significance of the different levels of convergence? I couldn't find a reference in the text.*

**We defined in the methods section that the GR diagnostic determines that the algorithm reaches convergence when the within-run variability ($\sigma_w$) is roughly equal to the between-run variability ($\sigma_b$), i.e. $\sigma_w/\sigma_b$ approaches 1. We considered that a model inversion had appropriately converged if the GR diagnostics was lower than 1.1 for each estimated parameter. See explanation in section 2.3.2 of the revised manuscript.**

*Minor point: section 4 (results and discussion) should actually be section 3.*

**The numbering was corrected.**

[revised manuscript text omitted]
 p | N p_wd | NSE p | NSE p_wd | KS p | KS p_wd | $E_{max}$ p | $E_{max}$ p_wd | $E_1$ | $s^*$ p | $s^*$ p_wd | $s_w$ p | $s_w$ p_wd |
|---|---|---|---|---|---|---|---|---|---|---|---|---|---|---|
| US-ARM (i) | point | 4 | 4 | 0.96 | 0.96 | 0.07 | 0.07 | 1.1 (11) | 1.3 (14) |  |  −0.7 (8) | 0.74 (5) | 0.19 (4) | 0.27 (7) |
|  |  | |  |  | |  |  | | | | | | | |
| (iii') | footprint | 3 | 3 | 0.94 | 0.94 | 0.08 | 0.06 | 1.7 (11) | 2 (12) | − | 0.62 (7) | −0.61 (9) | 24 (3) | 0.29 (2) |
|  |  | |  |  | |  | |  | | | | | | |
| (iv') | satellite | 3 | −3 | 0.96 | 97 | 0.08 | 0.09 | 0.7 (13) | 0.5 (13) | − | 0.42 (4) | 0.42 (4) | 0.24 (3) | 0.25 (2) |
| US-Ton | point | 3 | 4 | 0.95 | 0.97 | 0.09 | 0.08 | 2.3 (4) | 1.9 (10) | | 0.24 (6) | 0.33 (7) | 0.12 (1) | 0.18 (6) |
| | footprint | 3 | 3 | 0.94 | 0.98 | 0.13 | 0.08 | 2.2 (3) | 1.8 (8) | | 0.29 (2) | 0.4 (10) | 0.25 (0) | 0.26 (1) |
| | satellite | 3 | 9 | 0.99 | 0.99 | 0.06 | 0.07 | 1.2 (15) | 1 (13) | | 0.53 (12) | 0.62 (6) | 0.22 (3) | 0.26 (3) |
| US-MMS | point | 3 | 4 | 0.96 | 0.98 | 0.12 | 0.08 | 1.3 (3) | 1.1 (6) | | 0.34 (3) | 0.5 (8) | 0.29 (0) | 0.31 (2) |
| | footprint | 3 | 3 | 0.95 | 0.95 | 0.13 | 0.08 | 2.7 (6) | 4.5 (10) | | 0.82 (2) | 0.79 (3) | 0.38 (5) | 0.59 (1) |
| | satellite | 3 | 6 | 0.95 | 0.88 | 0.1 | 0.14 | 0.7 (8) | 0.9 (10) | | 0.65 (4) | 0.66 (3) | 0.28 (9) | 0.43 (2) |
| US-Me2 | point | 3 | 8 | 0.95 | 0.97 | 0.16 | 0.1 | 1.4 (3) | 1.1 (7) | | 0.33 (3) | 0.37 (8) | 0.29 (0) | 0.29 (1) |
| | footprint | 3 | 6 | 0.94 | 0.94 | 0.09 | 0.1 | 2.1 (2) | 2.9 (10) | | 0.23 (4) | 0.45 (5) | 0.15 (2) | 0.2 (6) |
|  | satellite | 3 | 4 | 0.89 | 0.89 | 0.12 | 0.1 | 1.6 (12) | 1.4 (15) | | 0.64 (8) | 0.66 (8) | 0.25 (3) | 0.31 (4) |

Values in parenthesis correspond to the coefficient of variation of the posterior parameter  percentage.
**p**, analytical model for the soil saturation pdf without seasons, $p_{wd}$, analytical model for the soil saturation pdf including wet and d of 20'000 simulation runs needed to obtain 3 converging results (see Sect. 2.3.2); NSE, quantile-level Nash Sutcliffe efficiency; KS, Kolmogorov Smirnov sta evapotranspiration in mm d$^{-1}$ (the weighted average wet and dry season $E_{max}$ is reported for the $p_{wd}$ model) ; $s^*$, point of incipient stomatal closure;   wilting point.